# Robust Manifold Estimation Approach for Evaluating Fidelity and Diversity

## Abstract

We propose a robust and reliable evaluation metric for generative models by introducing topological and statistical treatments for a rigorous support manifold estimation. Existing metrics, such as Inception Score (IS), Fréchet Inception Distance (FID), and the variants of `Precision` and `Recall` (`P&R`), heavily rely on support manifolds that are estimated from sample features. However, the reliability of their estimation has not been seriously discussed (and overlooked) even though the quality of the evaluation entirely depends on it. In this paper, we propose Topological Precision and Recall (`TopP&R`, pronounced "topper"), which provides a systematic approach to estimating support manifolds, retaining only topologically and statistically important features with a certain level of confidence. This not only makes `TopP&R` strong for noisy features, but also provides statistical consistency. Our theoretical and experimental results show that `TopP&R` is robust to outliers and non-independent and identically distributed (Non-IID) perturbations, while accurately capturing the true trend of change in samples. To the best of our knowledge, this is the first evaluation metric focused on the robust estimation of the support manifold and provides its statistical consistency under noise.

## 1 Introduction

In keeping with the remarkable improvements of deep generative models (Karras et al., 2019; 2020; 2021; Brock et al., 2018; Ho et al., 2020; Kingma & Welling, 2013; Sauer et al., 2022; 2021; Kang & Park, 2020), evaluation metrics that can well measure the performance of generative models have also been continuously developed (Salimans et al., 2016; Heusel et al., 2017; Sajjadi et al., 2018; Kynkäänniemi et al., 2019; Naeem et al., 2020). For instance, Inception Score (IS) (Salimans et al., 2016) measures the Kullback-Leibler divergence between the real and fake sample distributions. Fréchet Inception Score (FID)(Heusel et al., 2017) calculates the distance between the real and fake support manifolds using the estimated mean and variance under the multi-Gaussian assumption. The original Precision and Recall (Sajjadi et al., 2018) and its variants (Kynkäänniemi et al., 2019; Naeem et al., 2020) measure the fidelity and diversity by investigating whether the generated image belongs to the real image distribution and the generative model can reproduce all the real images in the distribution, respectively.

Considering the eminent progress of deep generative models based on these existing metrics, some may question why we need another evaluation study. In this paper, we argue that we need more reliable evaluation metrics now precisely, because deep generative models have reached sufficient maturity. To provide a more accurate and comprehensive ideas and to illuminate a new direction of improvements in the generative field, we need a more robust and reliable evaluation metric. In fact, it has been recently reported that even the most widely used evaluation metric, FID, sometimes doesn't match with the expected perceptual quality, fidelity, and diversity, which means the metrics are not always working properly (Kynkäänniemi et al., 2022). In addition to this, in practice, not only do generated samples but also real data in the wild often contain lots of artifacts (Pleiss et al., 2020; Li et al., 2022), and these have been shown to seriously perturb the existing evaluation metrics, giving a false sense of improvements (Naeem et al., 2020; Kynkäänniemi et al., 2022).

An ideal evaluation metric must capture the real signal of the data, while being robust to noise. Note that there is an inherent tension in developing metrics that meets these goals. On one hand, the metric should be sensitive enough so that it can capture real signals lurking in data. On the other hand, it

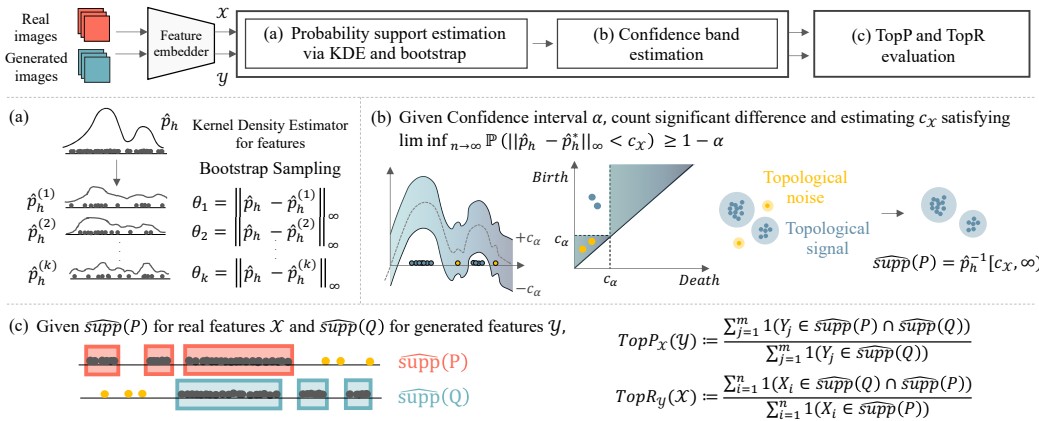

Figure 1: **Illustration of the proposed evaluation pipeline.** The proposed metric `TopP&R` is defined in the following three steps: (a) Confidence band estimation with bootstrapping in section 2, (b) Robust support estimation, and (c) Evaluation via `TopP&R` in section 3.

must ignore noises that hide the signal. However, sensitive metrics are inevitably susceptible to noise to some extent. To address this, one needs a systematic way to answer the following two questions: 1) what is signal and what is noise? and 2) how do we draw a line between them?

One solution can be to use the idea of statistical inference and topological data analysis (TDA). Topological data analysis (TDA) (Carlsson, 2009) is a recent and emerging field of data science that relies on topological tools to infer relevant features for possibly complex data. A key object in TDA is persistent homology, which observes how long each topological feature would survive over varying resolutions and provides a measure to quantify its significance; *i.e.*, if some features persist longer than others over varying resolutions, we consider them as topological signal and vice versa as noise.

In this paper, we propose to combine these ideas to form a more robust and compact feature manifold and overcome various issues from the conventional metrics. Our main contributions are as follows: we introduce **(1)** an approach to directly estimate a support manifold via Kernel Density Estimator (KDE) derived under topological conditions; **(2)** a new metric that is robust to outliers while reliably detecting the change of distributions on various scenarios; and **(3)** a theoretical guarantee of consistency with robustness under very weak assumptions that is suitable for high dimensional data; **(4)** combining a noise framework and a statistical inference in TDA—consistencies under noise framework have studied in much literature, but not quite in geometrical or topological setting.

## 2 BACKGROUND

To lay the foundation for our theoretical analysis, we introduce the main idea of persistent homology and its confidence estimation techniques that bring the benefit of using topological and statistical tools for addressing uncertainty in samples. In later sections, we use these tools to analyze the effects of outliers in evaluating generative models and provide more rigorous way of scoring the samples based on the confidence level we set. For space reasons, we only provide a brief overview of the concepts that are relevant to this work and refer the reader to Appendix A or (Edelsbrunner & Harer, 2010; Chazal & Michel, 2021; Wasserman, 2018; Hatcher, 2002) for further details.

### 2.1 NOTATION

For any $x$ and $r > 0$, we use the notation $\mathcal{B}_d(x, r) = \{y : d(y, x) < r\}$ be the open ball in distance $d$ of radius $r$. We also write $\mathcal{B}(x, r)$ when $d$ is understood from context. For a distribution $P$ on $\mathbb{R}^d$, we let $\text{supp}(P) := \{x \in \mathbb{R}^d : P(\mathcal{B}(x, r)) > 0 \text{ for all } r > 0\}$ be the support of $P$. Throughout the paper, we refer to $\text{supp}(P)$ as support manifold of $P$, or simply support, or manifold, but we don't necessarily require the (geometrical) manifold structure on $\text{supp}(P)$. For a kernel function $K : \mathbb{R}^d \to \mathbb{R}$, a dataset $\mathcal{X} = \{X_1, \ldots, X_n\} \subset \mathbb{R}^d$ and bandwidth $h > 0$, we let the kernel density estimator (KDE) as $\hat{p}_h(x) := \frac{1}{nh^d} \sum_{i=1}^{n} K\left(\frac{x - X_i}{h}\right)$, and we let the average KDE as $p_h := \mathbb{E}[\hat{p}_h]$.

We denote by $P, Q$ the probability distributions in $\mathbb{R}^d$ of real data and generated samples, respectively. And we use $\mathcal{X} = \{X_1, \ldots, X_n\} \subset \mathbb{R}^d$ and $\mathcal{Y} = \{Y_1, \ldots, Y_m\} \subset \mathbb{R}^d$ for real data and generated samples possibly with noise, respectively.

## 2.2 CONFIDENCE BAND ESTIMATION

Statistical inference has recently been developed for topological data analysis (Chazal et al., 2013; 2015; Fasy et al., 2014). Topological data analysis consists of features reflecting topological characteristics of data, and it is of question to distinguish features that are indeed from geometrical structures and features that are insignificant or due to noise. To statistically separate topologically significant features from topological noise, we use a confidence band. Given the significance level $\alpha$, let confidence band $c_\mathcal{X}$ be the bootstrap bandwidth of $\|\hat{p}_h - \hat{p}_h^*\|_\infty$. Then it satisfies $\liminf_{n\to\infty} \mathbb{P}\left(\|\hat{p}_h - p_h\|_\infty < c_\mathcal{X}\right) \geq 1 - \alpha$, as in Proposition 4 in Appendix C. This confidence band can be used to determine simultaneously significant topological features while filtering out noise features. The algorithm for computing $c_\mathcal{X}$ is described below.

---

**Algorithm 1** Confidence Band Estimator

---

1: *# KDE: kernel density estimator*
2: *# R.S.: random sample with replacement*
3: *# k: number of repeats*
4: *# $\hat{\theta}$: set of difference*
5: Given $\mathcal{X} = \{X_1, X_2, \ldots, X_n\}$
6: $\hat{p}_h = KDE(\mathcal{X})$
7: **for** *iteration* $= 1, 2, \ldots, k$ **do**
8:     *# compute $\hat{\theta}$ with bootstrap samples*
9:     $\mathcal{X}^* =$ R.S. $n$ times from $\mathcal{X}$
10:    *# $\hat{p}_h^*$ replaces population density*
11:    $\hat{p}_h^* = KDE(\mathcal{X}^*)$
12:    Append $\hat{\theta}$ with $\sqrt{n}\|\hat{p}_h - \hat{p}_h^*\|_\infty$
13: **end for**
14: *# grid search for the confidence band*

15: **for** $q \in [min(\hat{\theta}), max(\hat{\theta})]$ **do**
16:    count $= 0$
17:    **for** *element* $\in \hat{\theta}$ **do**
18:        *# count significant difference*
19:        **if** *element* $> q$ **then**
20:            $count = count + 1$
21:        **end if**
22:    **end for**
23:    *# define the band threshold*
24:    **if** $count/k \approx \alpha$ **then**
25:        $q_\alpha = q$
26:    **end if**
27: **end for**
28: *# define estimated confidence band*
29: $c_\alpha = q_\alpha/\sqrt{n}$

---

## 3 ROBUST SUPPORT MANIFOLD ESTIMATION FOR RELIABLE EVALUATION

Current evaluation metrics for generative models typically rely on strong regularity conditions. For example, they assume samples are well-curated without outliers or adversarial perturbation, real or generative models have bounded densities, etc. However, practical scenarios are wild: both real and generated samples can be corrupted with noise from various sources, and the real data can be very sparsely distributed without density.

In this work, we consider more general and practical situations, wherein both real and generated samples can have noises that come from sampling procedure, remained uncertainty due to data or model, etc. For more detailed discussions on the philosophy of our metric, please see Appendix E.

### 3.1 TOPOLOGICAL PRECISION AND RECALL

In the ideal case where we have full access to the probability distributions $P$ and $Q$, we define the precision and the recall of distributions as

$$\text{precision}_P(Q) := Q\left(\text{supp}(P)\right), \qquad \text{recall}_Q(P) := P\left(\text{supp}(Q)\right).$$

These correspond to the max precision and the max recall in Sajjadi et al. (2018). We tweak the precision as $\text{precision}_P(\mathcal{Y}) = Q\left(\text{supp}(P) \cap \text{supp}(Q)\right) / Q\left(\text{supp}(Q)\right)$, and define the precision of data points as

$$\text{precision}_P(\mathcal{Y}) := \frac{\sum_{j=1}^m \mathbf{1}\left(Y_j \in \text{supp}(P) \cap \text{supp}(Q)\right)}{\sum_{j=1}^m \mathbf{1}\left(Y_j \in \text{supp}(Q)\right)},$$

which is just replacing the distribution $Q$ by the empirical distribution $\frac{1}{m}\sum_{j=1}^{m}\delta_{Y_j}$ of $Y$ in the precision. We similarly define the recall of data points as

$$\text{recall}_Q(\mathcal{X}) := \frac{\sum_{i=1}^{n} 1\left(X_i \in \text{supp}(Q) \cap \text{supp}(P)\right)}{\sum_{i=1}^{n} 1\left(X_i \in \text{supp}(P)\right)},$$

However, in practice, $\text{supp}(P)$ and $\text{supp}(Q)$ are not known a priori and need to be estimated, and since we allow noise, these estimates should be robust to noise. For this, we use the kernel density estimator (KDE) and the bootstrap bandwidth to robustly estimate the support. Given $h_n > 0$ and a significance level $\alpha \in (0,1)$, we use the KDE $\hat{p}_{h_n}(x) := \frac{1}{nh_n^d}\sum_{i=1}^{n} K\left(\frac{x-X_i}{h_n}\right)$ of $\mathcal{X}$, and we use the bootstrap bandwidth $c_{\mathcal{X}}$ of $\left\|\hat{p}_{h_n} - \hat{p}_{h_n}^*\right\|_{\infty}$ from Section 2. Then we estimate the support of $P$ by the superlevel set at $c_{\mathcal{X}}$ as $\hat{\text{supp}}(P) = \hat{p}_{h_n}^{-1}[c_{\mathcal{X}}, \infty)$. Similarly, we let $\hat{q}_{h_m}(x) := \frac{1}{mh_m^d}\sum_{j=1}^{m} K\left(\frac{x-Y_j}{h_m}\right)$ be the KDE of $\mathcal{Y}$ and let $c_{\mathcal{Y}}$ be the bootstrap bandwidth of $\left\|\hat{q}_{h_m} - \hat{q}_{h_m}^*\right\|_{\infty}$, and then we use $\hat{\text{supp}}(Q) = \hat{q}_{h_m}^{-1}[c_{\mathcal{Y}}, \infty)$. Using the superlevel set at $c_{\mathcal{X}}$ allows to filter out noise whose KDE values are likely to be small.

For the robust estimates of the precision, we apply the support estimates to the precision of data points, and define the topological precision (TopP) as

$$\text{TopP}_{\mathcal{X}}(\mathcal{Y}) := \frac{\sum_{j=1}^{m} 1\left(Y_j \in \hat{\text{supp}}(P) \cap \hat{\text{supp}}(Q)\right)}{\sum_{j=1}^{m} 1\left(Y_j \in \hat{\text{supp}}(Q)\right)} = \frac{\sum_{j=1}^{m} 1\left(\hat{p}_{h_n}(Y_j) > c_{\mathcal{X}}, \ \hat{q}_{h_m}(Y_j) > c_{\mathcal{Y}}\right)}{\sum_{j=1}^{m} 1\left(\hat{q}_{h_m}(Y_j) > c_{\mathcal{Y}}\right)}.$$

And we similarly define the topological recall (TopR) as

$$\text{TopR}_{\mathcal{Y}}(\mathcal{X}) := \frac{\sum_{i=1}^{n} 1\left(\hat{q}_{h_m}(X_i) > c_{\mathcal{Y}}, \ \hat{p}_{h_n}(X_i) > c_{\mathcal{X}}\right)}{\sum_{i=1}^{n} 1\left(\hat{p}_{h_n}(X_i) > c_{\mathcal{X}}\right)}.$$

The kernel bandwidths $h_n$ and $h_m$ are hyperparameters that users need to choose. We also provide our guideline to select the optimal bandwidths $h_n$ and $h_m$ in practice. (See Appendix F.3)

### 3.2 BANDWIDTH ESTIMATION USING BOOTSTRAPPING

Using the bootstrap bandwidth $c_{\mathcal{X}}$ as threshold is the key part of our estimators TopP&R for robustly estimating $\text{supp}(P)$. As we have seen in Section 2, the bootstrap bandwidth $c_{\mathcal{X}}$ acts as a threshold for filtering out the topological noise in topological data analysis. Analogously, using $c_{\mathcal{X}}$ as a threshold allows to robustly estimating $\text{supp}(P)$. When $X_i$ is an outlier, its KDE value $\hat{p}_h(X_i)$ is likely to be small, and the KDE values at the connected component generated by $X_i$ is likely to be small as well. So those components from outliers are likely to be removed in the estimated support $\hat{p}_h^{-1}[c_{\mathcal{X}}, \infty)$. Higher dimensional homological noises are also removed. Hence, the estimated support denoises topological noise and robustly estimates $\text{supp}(P)$. See Appendix B for more detailed explanation.

Now that we are only left with topological features of high confidence, this allows us to draw analogies to confidence intervals in statistical analysis, where the uncertainty of the samples is treated by setting the level of confidence. In the next section, we show that TopP&R not only gives a more reliable evaluation score for generated samples but also has a good theoretical properties.

## 4 CONSISTENCY WITH ROBUSTNESS OF TopP&R

The key properties of TopP&R is consistency with robustness. The consistency ensures that, the precision and the recall we compute from the *data* approaches the precision and the recall from the *distribution* as we have more samples. The consistency allows to investigate the precision and the recall of the full distributions only with access to finite sampled data. TopP&R achieves consistency with robustness, that is, the consistency holds with the data possibly corrupted by noise. This is due to the robust estimation of the support with the kernel density estimator with confidence bands. This section is devoted to the theoretical analysis of consistency of TopP&R with robustness.

We demonstrate the statistical model for the data and the noise. Let $P, Q, \mathcal{X}, \mathcal{Y}$ be as in Notation in Section 2, and let $\mathcal{X}^0, \mathcal{Y}^0$ be real data and generated data without noise. $\mathcal{X}, \mathcal{Y}, \mathcal{X}^0, \mathcal{Y}^0$ are understood as multisets, *i.e.*, elements can be repeated. We first assume that the uncorrupted data are IID.

**Assumption 1.** *The data $\mathcal{X}^0 = \{X_1^0, \ldots, X_n^0\}$ and $\mathcal{Y}^0 = \{Y_1^0, \ldots, Y_m^0\}$ are IID from $P$ and $Q$, respectively.*

In practice, the data is often corrupted with noise. We consider the adversarial noise, where some fraction of data are replaced with arbitrary point cloud data.

**Assumption 2.** *Let $\{\rho_k\}_{k \in \mathbb{N}}$ be a sequence of nonnegative real numbers. Then the observed data $\mathcal{X}$ and $\mathcal{Y}$ satisfies $|\mathcal{X} \backslash \mathcal{X}^0| = n\rho_n$ and $|\mathcal{Y} \backslash \mathcal{Y}^0| = m\rho_m$.*

In the adversarial model, we control the level of noise by the fraction $\rho$, but do not assume other conditions such as IID or boundedness, to make our noise model very general and challenging.

For distributions and kernel functions, we assume weak condition, detailed in Assumption A1 and A2 in Appendix C. Under the data and the noise models, `TopP&R` achieves consistency with robustness. That is, the estimated precision and recall is asymptotically correct with high probability even if up to a portion of $1/\sqrt{n}$ or $1/\sqrt{m}$ are replaced by adversarial noise. This is due to the robust estimation of the support with the kernel density estimator with the confidence band of the persistent homology.

**Proposition 1.** *Suppose Assumption 1,2,A1,A2 hold. Suppose $h_n \to 0$, $nh_n \to \infty$, $nh_n^{-d}\rho_n^2 \to 0$, and similar relations hold for $h_m$, $\rho_m$. Then*

$$\left|\mathrm{TopP}_{\mathcal{X}}(\mathcal{Y}) - \mathrm{precision}_P(\mathcal{Y})\right| \to 0, \qquad \left|\mathrm{TopR}_{\mathcal{Y}}(\mathcal{X}) - \mathrm{recall}_Q(\mathcal{X})\right| \to 0, \qquad \textit{in probability.}$$

**Theorem 2.** *Under the same condition as in Proposition 1,*

$$\left|\mathrm{TopP}_{\mathcal{X}}(\mathcal{Y}) - \mathrm{precision}_P(Q)\right| \to 0, \qquad \left|\mathrm{TopR}_{\mathcal{Y}}(\mathcal{X}) - \mathrm{recall}_Q(P)\right| \to 0, \qquad \textit{in probability.}$$

Our theoretical results in Proposition 1 and Theorem 2 are novel and important in several perspetives. These results are among the first theoretical guarantees for evaluation metrics for generative models as far as we are aware of. Also, as in Remark 3, assumptions are very weak and suitable for high dimensional data. Also, robustness to adversarial noise is provably guaranteed.

## 5 EXPERIMENTS

A good evaluation metric must correctly capture the changes of the underlying data distribution. To examine the performance of evaluation metrics, we carefully select a set of experiments for sanity checks. With toy and real image data, we check 1) how well the metric captures the true trend of underlying data distributions and 2) how well the metric resist perturbations applied to samples. The shaded area of the figures denotes the $\pm 1$ standard deviation for ten trials.

### 5.1 SANITY CHECKS WITH TOY DATA

Following Naeem et al. (2020), we first examine how well the metric reflects the trend of $\mathcal{Y}$ moving away from $\mathcal{X}$ and whether it is suitable for finding mode-drop phenomena. In addition to these, we newly design several experiments that can highlight `TopP&R`'s favorable theoretical properties of consistency with robustness in various scenarios.

#### 5.1.1 SHIFTING THE GENERATED FEATURE MANIFOLD

For this experiment, we generate samples for $\mathcal{X} \sim \mathcal{N}(0, I)$ and $\mathcal{Y} \sim \mathcal{N}(\mu\mathbf{1}, I)$ in $\mathbb{R}^{64}$ where $\mathbf{1}$ is a vector of ones and $I$ is an identity matrix. We then examine how each metric responds to shifting $\mathcal{Y}$ with $\mu \in [-1, 1]$ while there are outliers at $\mathbf{3} \in \mathbb{R}^{64}$ for both $\mathcal{X}$ and $\mathcal{Y}$ (Figure 2). Here, we find that both improved `P&R` and `D&C` behave pathologically when there are outliers. Since these methods are based on the k-nearest neighbor algorithm and ignore the fact that there can be outliers in both real and fake data, they inevitably overestimate the underlying support when there are outliers. For example, when $\mathcal{X}$ lies between $\mathcal{Y}$ and the outlier at $y = \mathbf{3}$, `Recall` returns a high-diversity score, even though the true supports of $\mathcal{X}$ and $\mathcal{Y}$ are actually far apart. In addition, `P&R` does not reach 1 in high dimensions even when $\mathcal{X} = \mathcal{Y}$. Naeem et al. (2020) circumvented these problems by proposing `D&C` that always use $\mathcal{X}$ (the real data distribution) as a reference point, which in most cases is assumed to have fewer outliers than $\mathcal{Y}$ (the fake data distribution). However, there is no guarantee that this will be the case in practice. When there is an outlier in $\mathcal{X}$, `D&C` also returns an incorrect high-fidelity score at $\mu > 0.5$. On the other hand, `TopP&R` shows a stable trend unaffected by outliers, demonstrating the robustness of our method.

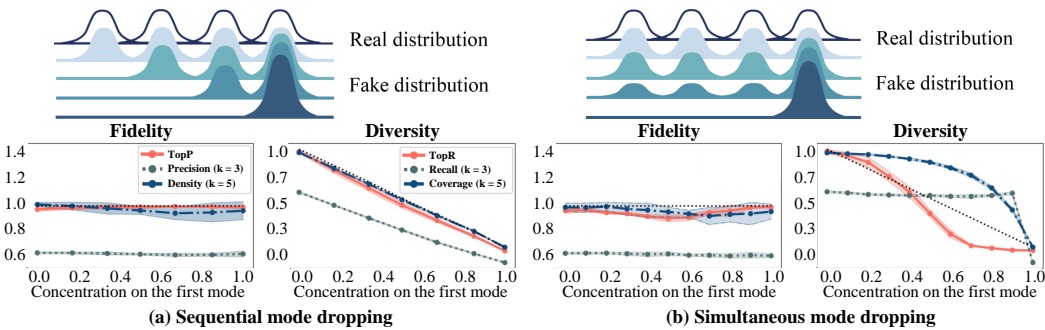

Figure 2: Behaviors of evaluation metrics for outliers on real and fake distribution. The horizontal axis corresponds to the value of $\mu$.

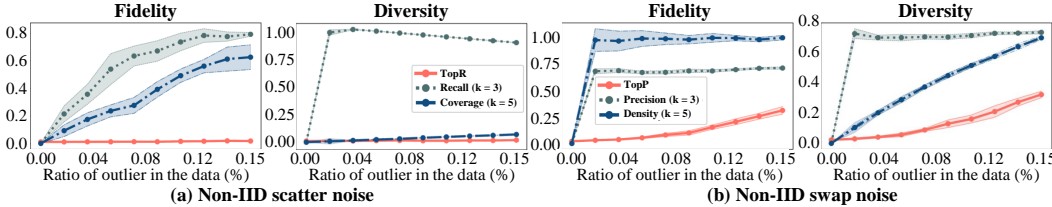

Figure 3: Behaviors of evaluation metrics for (a) sequential and (b) simultaneous mode dropping scenarios. The horizontal axis shows the concentration ratio on the distribution centered at $\mu = 0$.

Figure 4: Behaviors of evaluation metrics on Non-IID perturbations. We replace a certain percentage of real and fake data (a) with random uniform noise and (b) by switching.

### 5.1.2 SEQUENTIALLY AND SIMULTANEOUSLY DROPPING MODES

For this experiment, we consider the mixture of Gaussians with seven modes in $\mathbb{R}^{64}$. We simulate mode-drop phenomena by gradually dropping all but one mode from the fake distribution $\mathcal{Y}$ that is initially identical to $\mathcal{X}$ (Figure 3). As in the illustration of mode-drop experiment, when the number of samples in a particular mode decreases, we kept the number of samples in $\mathcal{X}$ constant so that the same amount of decreased samples are supplemented to the first mode which leads fidelity to be fixed to 1. From the result, we observe that the values of `Precision` fail to saturate, *i.e.*, mainly smaller than 1, and the `Density` fluctuates to a value greater than 1 indicating their instability and unboundedness. In terms of diversity, `Recall` does not respond to the simultaneous mode drop, nor does the improved metric `Coverage` show a fast decay as the reference line. Compared to these methods, `TopP` performs well, being held at the upperbound of 1 in sequential mode dropping, and `TopR` also decreases closest to the reference line in simultaneous mode drops.

### 5.1.3 TOLERANCE TO NON-IID PERTURBATIONS

Robustness to perturbations is another important aspect we should consider when designing a metric. Here, we test whether `TopP&R` behaves stably under two variants of noise cases; 1) **scatter noise**: replacing $X_i$ and $Y_j$ with uniformly distributed noise and 2) **swap noise**: swapping the position between $X_i$ and $Y_j$. These two cases all correspond to the adversarial noise model of Assumption 2. We set $\mathcal{X} \sim \mathcal{N}(\mu = 0, I) \in \mathbb{R}^{64}$ and $\mathcal{Y} \sim \mathcal{N}(\mu = 1, I) \in \mathbb{R}^{64}$ where $\mu = 1$, and thus an ideal evaluation metric must return zero for both fidelity and diversity. In both cases, we find that `P&R` and `D&C` are more sensitive while `TopP&R` remains relatively stable until the noise ratio reaches 15% of the total data, which is a clear example of the weakness of existing metrics to perturbation (Figure 4).

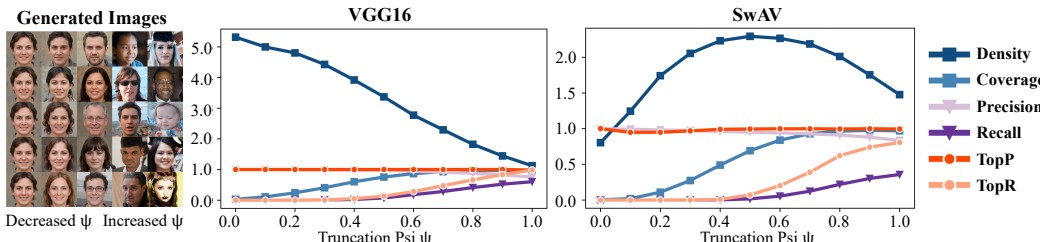

Figure 5: Behaviour of metrics with truncation trick. The horizontal axis corresponds to the value of $\psi$ denoting the increased diversity. The images are generated via StyleGAN2 with FFHQ dataset.

## 5.2 SANITY CHECK WITH REAL DATA

Now that we have verified the metrics on toy data using Gaussians, we test them on real data. Just like in the toy experiments, we concentrate on how the metrics behave in extreme situations, such as outliers, mode-drop phenomena, perceptual distortions, and etc. We also test the image embedder for evaluation, including pretrained VGG16 (Simonyan & Zisserman, 2014), InceptionV3 (Szegedy et al., 2016), and SwAV (Morozov et al., 2020). Here, linear random projection to 32 dimension is additionally used for `TopP&R`. For more experimental details, please refer to the Appendix F.1.

### 5.2.1 RESOLVING FIDELITY AND DIVERSITY

To test whether `TopP&R` responds appropriately to the change in the underlying distributions in real scenarios, we test the metric on the generated images of StyleGAN2 (Karras et al., 2020) using the truncation trick (Karras et al., 2019). As shown in Figure 5, every time the distribution is transformed by $\psi$, `TopP&R` responds well and shows consistent behavior across different embedders with bounded scores in $[0, 1]$, which are important virtues as an evaluation metric. On the other hand, `Density` gives unbounded scores (fidelity > 1) and shows inconsistent trend depending on the embedder. Because `Density` is not capped in value, it is difficult to interpret the score and know exactly which value denotes the best performance (*e.g.*, in our case, the best performance is when `TopP&R` = 1). Since `TopP&R` pays more attention to the consistent behavior of a model by examining what the model primarily generates, rather than relying on the entire sample, which contains results by chance, the fact that `TopP` is kept at 1.0 means that StyleGAN2 produces high-quality images most of the time. Thus, this behavior ("`TopP` remains constant") does not mean that `TopP` is inferior to regular precision for checking the trade-off between fidelity and diversity, but rather reveals its property focusing on different perspectives than the others.

### 5.2.2 SEQUENTIALLY AND SIMULTANEOUSLY DROPPING MODES IN CIFAR-10

We conduct an additional simultaneous mode drop experiment to verify `TopP&R`'s actual sensitiveness on the real data set (CIFAR-10). The performance of each metric (Figure A2) is measured with the identical data while simultaneously dropping the modes of nine classes of CIFAR-10. Since the number of the images dropped in each step is identical, the trend of ground truth diversity should linearly decrease. Here, `P&R` metric captures the simultaneous mode dropping better than `D&C` because this time random drop of the modes has reduced the area of the estimated fake manifold. On the other hand, `TopP&R` best captures the true trend of decreasing diversity on average, consistent with the toy result in Figure 3. In addition, we perform the experiments on a dataset with long-tailed distribution and find that `TopP&R` captures the trend well even when there are minority sets (Appendix G.2). This again shows the reliability of `TopP&R`.

### 5.2.3 ROBUSTNESS TO PERTURBATIONS BY OUTLYING FEATURES

To demonstrate the robustness of our metric against the adversarial noise model of Assumption 2, we test both scatter-noise and swap noise scenarios with real data. In the experiment, following Kynkäänniemi et al. (2019), we first classify inliers and outliers that are generated by StyleGAN (Karras et al., 2019). For scatter noise we add the outliers to the inliers and for swap noise we swap the real FFHQ images with generated images. Under these specific noise conditions, `Precision`

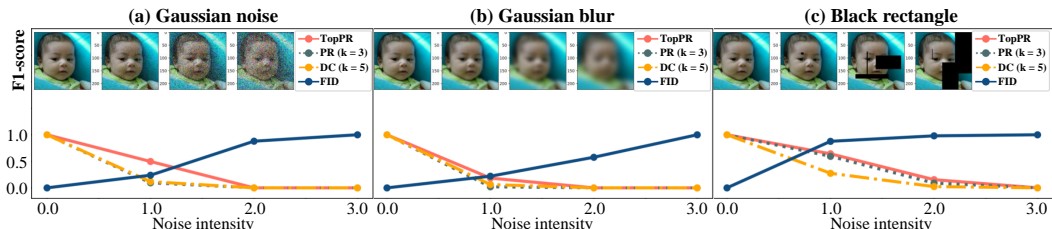

Figure 6: Comparison of evaluation metrics on Non-IID perturbations using FFHQ dataset. We replaced certain ratio of $\mathcal{X}$ and $\mathcal{Y}$ (a) with outliers and (b) by exchanging features.

Figure 7: Verification of whether `TopP&R` can make an accurate quantitative assessment of noisy image features. Gaussian Noise, gaussian blur, and black rectangle noise are added on the FFHQ imageset and embedded with T4096.

shows similar or even better robustness than `Density` (Figure 6). On the other hand, `Coverage` is more robust than `Recall`. In both cases, `TopP&R` shows the best performance, resistant to noise.

### 5.2.4 SENSITIVENESS TO THE NOISE INTENSITY

One of the advantages of FID (Heusel et al., 2017) is that it is good at estimating the degrees of distortion applied to the images. Similarly, we check whether the F1-score based on `TopP&R` provides a reasonable evaluation according to different noise levels. As illustrated in Figure 7, $\mathcal{X}$ and $\mathcal{Y}$ are sets of reference FFHQ features and noisy FFHQ features, respectively. The experimental results show that `TopP&R` actually reflects well the different degrees of distortion added to the images.

### 5.2.5 RANKING BETWEEN GENERATIVE MODELS

One of the major caveats with two-score metrics is that they make it difficult to rank between different models; *e.g.*, which model is better? High fidelity with low diversity or low fidelity with high diversity? In the case of traditional precision and Recall, this problem could be solved by using F1-score, which is the harmonic mean of fidelity and diversity. However, unlike the traditional ones, the F1-score based on `P&R` or `D&C` does not provide a reliable or stable score due to their inherent instability and unboundedness. Thanks to its stability and robustness to various perturbations, we find that the `TopP&R`-based F1 score offers consistent ranking with FID under various embedding networks (Table 1)[1]. To quantitatively compare between the similarity of rankings across varying embedders by different metrics, we have computed Hamming Distance (HD) (Appendix F.4) where lower HD indicates more similarity. `TopP&R`, `P&R`, and `D&C` have HDs of 1.33, 2.66, and 3.0, respectively. From this, `TopP&R` provides the most consistent ranking across varying embedders (consistent to Section 5.2.1).

## 6 RELATED WORKS

**Persistent homology and deep learning.** Topology shows various potentials in the field of deep learning by introducing a new perspective on the support estimation, a new distance function robust to the noisy information, and a technique for GAN evaluation. Chen et al. (2017) introduces a method for

---

[1]All GAN models used in the experiment follow the settings in StudioGAN PyTorch-StudioGAN is an open-source library under the MIT license (MIT), which are under the NVIDIA source code license.

| | Model | StyleGAN2 | ReACGAN | BigGAN | PDGAN | ACGAN | WGAN-GP |
|---|---|---|---|---|---|---|---|
| **InceptionV3** | **FID** (↓) | 3.78 (1) | 3.87 (2) | 4.16 (3) | 31.54 (4) | 33.39 (5) | 107.68 (6) |
| | **TopP&R** (↑) | 0.9769 (1) | 0.8457 (2) | 0.7751 (3) | 0.7339 (4) | 0.6951 (5) | 0.0163 (6) |
| | D&C (↑) | 0.9626 (2) | 0.9409 (3) | 1.1562 (1) | 0.4383 (4) | 0.3883 (5) | 0.1913 (6) |
| | P&R (↑) | 0.6232 (1) | 0.3320 (2) | 0.3278 (3) | 0.1801 (4) | 0.0986 (5) | 0.0604 (6) |
| **VGG16** | **TopP&R** (↑) | 0.9754 (1) | 0.5727 (3) | 0.7556 (2) | 0.4021 (4) | 0.3463 (5) | 0.0011 (6) |
| | D&C (↑) | 0.9831 (3) | 1.0484 (1) | 0.9701 (4) | 0.9872 (2) | 0.8971 (5) | 0.6372 (6) |
| | P&R (↑) | 0.6861 (1) | 0.1915 (3) | 0.3526 (2) | 0.0379 (4) | 0.0195 (5) | 0.0001 (6) |
| **SwAV** | **TopP&R** (↑) | 0.9093 (1) | 0.3568 (3) | 0.5578 (2) | 0.1592 (4) | 0.1065 (5) | 0.0003 (6) |
| | D&C (↑) | 1.0732 (1) | 0.9492 (3) | 1.0419 (2) | 0.6328 (4) | 0.4565 (5) | 0.0721 (6) |
| | P&R (↑) | 0.5623 (1) | 0.0901 (3) | 0.1459 (2) | 0.0025 (4) | 0.0000 (6) | 0.0002 (5) |

Table 1: Generative models ranked by FID and F1-scores based on `TopP&R`, `D&C`, and `P&R`, respectively. The $\mathcal{X}$ and $\mathcal{Y}$ are embedded with InceptionV3, VGG16, and SwAV. The number inside the parenthesis denotes the rank based on each metric.

approximating the support of a distribution using general density estimator and the Hausdorff distance and a new visualization method for support. Chazal et al. (2011) proposes distance-to-measure, a robust Wasserstein distance function for perturbation, as an alternative to the characteristic that existing distance functions are not robust to outliers. For the evaluation, one of the recent metric called MTop-Divergence (Barannikov et al., 2021) uses the summation (or in another word statistics) of the life-length of homology to score which manifold is containing more important topological signals. While MTop-Divergence directly use persistent homology to score the deep-learning models, we employ topology to estimate a robust and stable manifold.

**Evaluation metrics.** Various evaluation metrics for generative models have been recently proposed (Salimans et al., 2016; Heusel et al., 2017; Sajjadi et al., 2018; Kynkäänniemi et al., 2019; Naeem et al., 2020; Borji, 2022). One of the earliest methods is Inception Score (IS) (Szegedy et al., 2016), which measures the divergence of generated samples on the InceptionV3 embedding space. However, IS fails to capture the simultaneous mode drop and only considers the population distribution. Fréchet Inception Distance (FID) (Heusel et al., 2017) measures the difference in the means and variances of the real and fake features. Since FID assumes the multi-Gaussian distribution of the features, if the true feature distribution is not normally distributed, the estimation becomes highly unreliable. Unlike IS and FID, which give a single score, some metrics separate the score into two components, the fidelity and diversity Sajjadi et al. (2018); Kynkäänniemi et al. (2019); Naeem et al. (2020). While Topological Precision and Recall (`TopP&R`) falls into this category, unlike the others, it does not assume strong regularity conditions.

# 7 CONCLUSIONS

Recently, many works have been proposed to score the fidelity and diversity of generative models. However, none of them has focused on an accurate estimation of supports even though this is one of the key components in the evaluation pipeline. In this paper, we proposed topological precision and recall (`TopP&R`) that provides a systematical fix for robustly estimating the manifold by employing topological and statistical ideas. Our theoretical and experimental results showed that `TopP&R` serves as a robust and reliable evaluation metric under various embeddings and noisy conditions, including mode collapse, outliers, and Non-IID perturbations.

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

APPENDIX

## A  MORE BACKGROUND ON TOPOLOGICAL DATA ANALYSIS

Topological data analysis (TDA) (Carlsson, 2009) is a recent and emerging field of data science that relies on topological tools to infer relevant features for possibly complex data. A key object in TDA is persistent homology, which quantifies salient topological features of data by observing them in multi-resolutions.

**Filtration.** A *filtration* is a collection of subspaces approximating the data points at different resolutions, formally $\mathcal{F} = \{\mathcal{F}_\delta \subset \mathbb{R}^d\}_{\delta \in \mathbb{R}}$ such that $\delta_1 \leq \delta_2$ implies that $\mathcal{F}_{\delta_1} \subset \mathcal{F}_{\delta_2}$. Typically, a filtration is defined through a function $f$ related to data. Given a function $f \colon \mathbb{R}^d \to \mathbb{R}$, we consider its sublevel filtration $\{f^{-1}(-\infty, \delta]\}_{\delta \in \mathbb{R}}$ or a superlevel filtration $\{f^{-1}[\delta, \infty)\}_{\delta \in \mathbb{R}}$.

**Persistent homology.** *Persistent homology* is a multiscale approach to represent the topological features, and is represented in the persistence diagram. For a filtration $\mathcal{F}$ and for each nonnegative $k$, we track when $k$-dimensional homological features (e.g., 0-dimension: connected component, 1-dimension: loop, 2-dimension: cavity,...) appear and disappear in the filtration. As increasing or decreasing $\delta$ in the filtration $\{\mathcal{F}_\delta\}$, if a homological feature appears at $F_b$ and disappears at $F_d$, then we say that it is born at $b$ and dies at $d$. By considering these pairs $\{(b, d)\}$ as points in the plane $(\mathbb{R} \cup \{\pm\infty\})^2$, we obtain a *persistence diagram*. From this, a homological feature with a longer life length, $d - b$, can be treated as a significant feature in the data set, and a homological feature with a shorter life length as a topological noise, which lies near the diagonal line $\{(\delta, \delta) : \delta \in \mathbb{R}\}$.

As discussed above, a homological feature with a long life-length is an important information in topology while the homology with short life-length can be treated as a non-significant information or noise. The confidence band estimator provides the confidence set from the features that only includes topologically and statistically significant (statistically considered as elements in the population set) under a certain level of confidence. One way of constructing the confidence set uses the superlevel filtration of kernel density estimator and the bootstrap confidence band. Let $\mathcal{X} = \{X_1, X_2, ..., X_n\}$ as given points cloud, then the probability for the distribution of points can be estimated via KDE defined as following: $\hat{p}_h(x) := \frac{1}{nh^d} \sum_{i=1}^n K(\frac{x - X_i}{h})$ where h is the bandwidth and d as a dimension of the space. We derive estimated likelihood of $\mathcal{X}$ with KDE and likehood of $\tilde{p}$ with using bootstrapped samples $\mathcal{X}^*$. Now, given the significance level $\alpha$ and $h > 0$, let confidence band $q_\mathcal{X}$ be bootstrap bandwidth of a Gaussian Empirical Process (van der Vaart, 2000; Kosorok, 2008), $\sqrt{n}||\hat{p}_h - \hat{p}_h^*||_\infty$. Then it satisfies $P(\sqrt{n}||\hat{p}_h - p_h||_\infty < q_\mathcal{X}) \geq 1 - \alpha$, as in Proposition 4 in Section C. Then the ball of persistent homology centered at $\hat{\mathcal{P}}_h$ and radius $c_\mathcal{X} = q_\mathcal{X}/\sqrt{n}$ in the bottleneck distance $d_B$ is a valid confidence set as $\liminf_{n \to \infty} \mathbb{P}\left(\mathcal{P} \in \mathcal{B}_{d_B}(\hat{\mathcal{P}}_h, c_\mathcal{X})\right) \geq 1 - \alpha$. This confidence set has further interpretation that in the persistence diagram, homological features that are above twice the radius $2c_\mathcal{X}$ from the diagonal are simultaneously statistically significant.

## B  DENOISING TOPOLOGICAL FEATURES FROM OUTLIERS

Using the bootstrap bandwidth $c_\mathcal{X}$ as threshold is the key part of our estimators TopP&R for robustly estimating supp($P$). When the level set $\hat{p}_h^{-1}[c_\mathcal{X}, \infty)$ is used, the homology of $\hat{p}_h^{-1}[c_\mathcal{X}, \infty)$ consists of homological features whose (birth) $\geq c_\mathcal{X}$ and (death) $\leq c_\mathcal{X}$, which are the homological features in skyblue area in Figure A1. In this example, we conisder three types of homological noise, though there can be many more corresponding to different homological dimensions.

- There can be a 0-dimensional homological noise of (birth) $< c_\mathcal{X}$ and (death) $< c_\mathcal{X}$, which is the red point in the persistence diagram of Figure A1. This noise corresponds to the orange connected component on the left. As in the figure, this type of homological noise usually corresponds to outliers.

- There can be a 0-dimensional homological noise of (birth) $> c_\mathcal{X}$ and (death) $> c_\mathcal{X}$, which is the green point in the persistence diagram of Figure A1. This noise corresponds to the connected component surrounded by green line on the left. As in the figure, this type of homological noise lies within the estimated support, not like the other two.

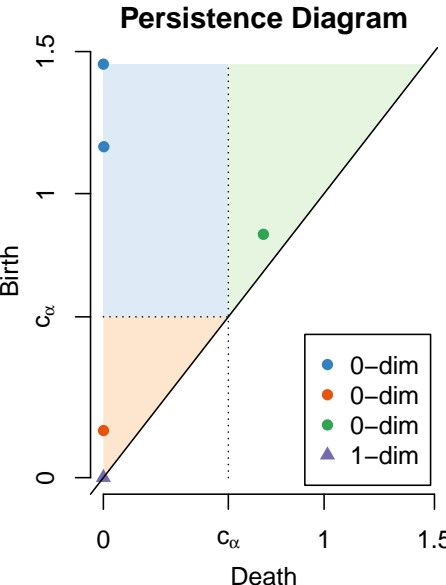

Figure A1: To robustly estimate the support, we use the bootstrap bandwidth $c_\alpha$ to filter out topological noise (orange) and keep topological signal (skyblue). Then `TopP&R` is computed on this support.

- There can be a 1-dimensional homological noise of $(\text{birth}) < c_\mathcal{X}$ and $(\text{death}) < c_\mathcal{X}$, which is the purple point in the persistence diagram of Figure A1. This noise corresponds to the purple loop on the left.

These homological noises satisfy either their $(\text{birth}) < c_\mathcal{X}$ and $(\text{death}) < c_\mathcal{X}$ or their $(\text{birth}) > c_\mathcal{X}$ and $(\text{death}) > c_\mathcal{X}$ simultaneously with high probability, so those homological noises are removed in the estimated support $\hat{p}_h^{-1}[c_\mathcal{X}, \infty)$, which is the blue area in the left and the skyblue area in the right in Figure A1.

We would like to further emphasize that homological noises are not restricted to 0-dimension lying outside the estimated support (red point in the persistence diagram of Figure A1). 0-dimensional homological noise inside the estimated support (green point in the persistence diagram of Figure A1), 1-dimensional homological noise can also arise, and the bootstrap bandwidth $c_\mathcal{X}$ allows to simultaneously filter them.

## C  ASSUMPTIONS ON DISTRIBUTIONS AND KERNELS

For distributions, we assume that the order of probability volume decay $P(\mathcal{B}(x,r))$ is at least $r^d$.

**Assumption A1.** *For all $x \in \text{supp}(P)$ and $y \in \text{supp}(Q)$,*

$$\liminf_{r \to 0} \frac{P(\mathcal{B}(x,r))}{r^d} > 0, \qquad \liminf_{r \to 0} \frac{Q(\mathcal{B}(y,r))}{r^d} > 0.$$

*Remark* 3. Assumption A1 is analogous to Assumption 2 of Kim et al. (2019), but is weaker since the condition is pointwise on each $x \in \mathbb{R}^d$. And this condition is much weaker than assuming a density on $\mathbb{R}^d$: for example, a distribution supported on a low-dimensional manifold satisfies Assumption A1. This provides a framework suitable for high dimensional data, since many times high dimensional data lies on a low dimensional structure hence its density on $\mathbb{R}^d$ cannot exist. See Kim et al. (2019) for more detailed discussion.

For kernel functions, we assume the following regularity conditions:

**Assumption A2.** *Let $K : \mathbb{R}^d \to \mathbb{R}$ be a nonnegative function with $\|K\|_1 = 1$, $\|K\|_\infty, \|K\|_2 < \infty$, and satisfy the following:*

*(1)* $K(0) > 0$.

*(2)* $K$ *has a compact support.*

*(3)* $K$ *is Lipschitz continuous and of second order.*

Assumption A2, allows to build a valid bootstrap confidence band for kernel density estimator (KDE). See Theorem 12 of (Fasy et al., 2014) or Theorem 3.4 of (Neumann, 1998)

**Proposition 4** (Theorem 3.4 of (Neumann, 1998))**.** *Let* $\mathcal{X} = \{X_1, \ldots, X_n\}$ *be IID from a distribution* $P$. *For* $h > 0$, *let* $\hat{p}_h, \hat{p}_h^*$ *be kernel density estimator for* $\mathcal{X}$ *and its bootstrap* $\mathcal{X}^*$, *respectrively, and for* $\alpha \in (0, 1)$, *let* $c_{\mathcal{X}}$ *be the* $\alpha$ *bootstrap quantile from* $\sqrt{nh^d} \|\hat{p}_h - \hat{p}_h^*\|_\infty$. *For* $h_n \to 0$,

$$\mathbb{P}\left(\sqrt{nh_n^d} \|\hat{p}_{h_n} - p_{h_n}\|_\infty > c_{\mathcal{X}}\right) = \alpha + \left(\frac{\log n}{nh_n^d}\right)^{\frac{4+d}{4+2d}}.$$

Assumption A1, A2 ensures that, when the bandwidth $h_n \to 0$, average KDEs are bounded away from 0.

**Lemma 5.** *Let* $P$ *be a distribution satisfying Assumption A1. Suppose* $K$ *is a nonnegative function satisfying* $K(0) > 0$ *and continuous at* $0$. *Suppose* $\{h_n\}_{n \in \mathbb{N}}$ *with* $h_n \geq 0$ *and* $h_n \to 0$ *is given. Then for all* $x \in \text{supp}(P)$,

$$\liminf_n p_{h_n}(x) > 0.$$

*Proof.* Since $K(0) > 0$ and $K$ is continuous at 0, there is $r_0 > 0$ such that for all $y \in B(0, r_0)$, $K(y) \geq \frac{1}{2}K(0) > 0$. And hence

$$p_h(x) = \int \frac{1}{h^d} K\left(\frac{x-y}{h}\right) dP(y) \geq \int \frac{K(0)}{2h^d} 1\left(\frac{x-y}{h} \in B(0, r_0)\right) dP(y)$$
$$\geq \frac{K(0)}{2h^d} P\left(B(x, r_0 h)\right).$$

Hence as $h_n \to 0$,

$$\liminf_n p_{h_n}(x) > 0.$$

$\square$

# D    DETAILS AND PROOFS FOR SECTION 4

Let $\tilde{p}_h$ be the KDE on $\mathcal{X}^0$. For a finite set $\mathcal{X}$, we use the notation $c_{\mathcal{X},\alpha}$ for $\alpha$-bootstrap quantile satisfying $\mathbb{P}\left(\|\hat{p}_{\mathcal{X},h_n} - \hat{p}_{\mathcal{X}^b,h_n}\|_\infty > c_{\mathcal{X},\alpha} | \mathcal{X}\right) = 1 - \alpha$, where $\mathcal{X}^b$ is the bootstrap sample from $\mathcal{X}$. For a distribution $P$, we use the notation $c_{P,\alpha}$ for $\alpha$-quantile satisfying $\mathbb{P}\left(\|\hat{p}_{h_n} - p_{h_n}\|_\infty > c_{P,\alpha}\right) = 1 - \alpha$, where $\hat{p}_{h_n}$ is kernel density estimator of IID samples from $P$. Hence when $\mathcal{X}$ is not IID samples from $P$, the relation of Proposition 4 may not hold.

**Lemma 6.**        *(1)* *Under Assumption 1, 2 and A2,*

$$\|\hat{p}_h - \tilde{p}_h\|_\infty \leq \frac{\rho_n \|K\|_\infty}{h^d}.$$

*(2)* *Under Assumption 1, 2 and A2,*

$$c_{\mathcal{X}^0, \alpha+\delta} - O\left(\rho_n + \sqrt{\frac{\rho_n \log(1/\delta)}{nh^{2d}}}\right) \leq c_{\mathcal{X},\alpha} \leq c_{\mathcal{X}^0, \alpha-\delta} + O\left(\rho_n + \sqrt{\frac{\rho_n \log(1/\delta)}{nh^{2d}}}\right).$$

*(3)* *Suppose Assumption 1, 2,A2 hold, and suppose* $nh_n^{-d}\rho_n^2 \to 0$. *Then with probability* $1 - \alpha - 2\delta$,

$$\|\hat{p}_h - p_h\|_\infty < c_{\mathcal{X},\alpha} \leq c_{P,\alpha-\delta}.$$

*Proof.* (1)

First, note that

$$\hat{p}_h - \tilde{p}_h = \frac{1}{nh^d} \sum_{i=1}^{n} \left( K\left(\frac{x - X_i}{h}\right) - K\left(\frac{x - X_i^0}{h}\right) \right).$$

Then under Assumption A2,

$$\|\hat{p}_h - \tilde{p}_h\|_\infty \le \frac{1}{nh^d} \sum_{i=1}^{n} \left\| K\left(\frac{x - X_i}{h}\right) - K\left(\frac{x - X_i^0}{h}\right) \right\|_\infty$$

$$\le \frac{1}{nh^d} \sum_{i=1}^{n} \|K\|_\infty I\left(X_i \ne X_i^0\right).$$

Then from Assumption 2, $\sum_{i=1}^{n} I\left(X_i \ne X_i^0\right) \le n\rho_n$, and hence

$$\|\hat{p}_h - \tilde{p}_h\|_\infty \le \frac{\|K\|_\infty \rho_n}{h^d}.$$

(2)

Let $\mathcal{X}_b$, $\mathcal{X}_b^0$ be bootstrapped samples of $\mathcal{X}$, $\mathcal{X}^0$ with the same sampling with replacement process. Let $\hat{p}_h^b, \tilde{p}_h^b$ be KDE of $\mathcal{X}_b$ and $\mathcal{X}_b^0$, respectively. And, note that

$$\left| \left\|\hat{p}_h - \hat{p}_h^b\right\|_\infty - \left\|\tilde{p}_h - \tilde{p}_h^b\right\|_\infty \right| \le \|\hat{p}_h - \tilde{p}_h\|_\infty + \left\|\hat{p}_h^b - \tilde{p}_h^b\right\|_\infty.$$

Let $L_b$ be the number of elements where $\mathcal{X}_b$ and $\mathcal{X}_b^0$ differ, i.e., $L_b = \left|\mathcal{X}_b \backslash \mathcal{X}_b^0\right| = \left|\mathcal{X}_b^0 \backslash \mathcal{X}_b\right|$, then $L_b \sim \text{Binomial}(n, \rho_n)$, and

$$\left\|\hat{p}_h^b - \tilde{p}_h^b\right\|_\infty \le \frac{\|K\|_\infty L_b}{nh^d}.$$

And hence,

$$\left| \left\|\hat{p}_h - \hat{p}_h^b\right\|_\infty - \left\|\tilde{p}_h - \tilde{p}_h^b\right\|_\infty \right| \le \frac{\|K\|_\infty (n\rho_n + L_b)}{nh^d}.$$

Then by using subgaussian tail bound, with probability $1 - \delta$,

$$\left| \left\|\hat{p}_h - \hat{p}_h^b\right\|_\infty - \left\|\tilde{p}_h - \tilde{p}_h^b\right\|_\infty \right| \le O\left( \rho_n + \sqrt{\frac{\rho_n \log(1/\delta)}{nh^{2d}}} \right).$$

Hence this implies

$$c_{\mathcal{X}^0, \alpha+\delta} - O\left( \frac{\rho_n}{h^d} + \sqrt{\frac{\rho_n \log(1/\delta)}{nh^{2d}}} \right) \le c_{\mathcal{X}, \alpha} \le c_{\mathcal{X}^0, \alpha-\delta} + O\left( \frac{\rho_n}{h^d} + \sqrt{\frac{\rho_n \log(1/\delta)}{nh^{2d}}} \right).$$

(3)

Since $\mathcal{X}^0$ is IID samples from $P$, with probability $1 - \alpha - 2\delta$,

$$\|\tilde{p}_h - p_h\|_\infty < c_{P, \alpha+2\delta} \le c_{\mathcal{X}^0, \alpha+2\delta} + O\left(\frac{1}{nh^d}\right).$$

Now, note that $c_{\mathcal{X}^0, \alpha} = \Theta\left( \sqrt{\frac{\log(1/\alpha)}{nh^d}} \right)$, and hence

$$c_{\mathcal{X}^0, \alpha} - c_{\mathcal{X}^0, \alpha+\delta} = \Theta\left( \sqrt{\frac{\log(1/\alpha)}{nh^d}} - \sqrt{\frac{\log(1/(\alpha+\delta))}{nh^d}} \right) \ge \Omega\left( \frac{\log((\alpha+\delta)/\alpha)}{\sqrt{nh^d}} \right).$$

Then under Assumption 2, since $nh_n^{-d}\rho_n^2 = o(1)$ and $h_n^{-d}\rho_n = o(1)$,

$$\|\hat{p}_h - p_h\| \le \|\tilde{p}_h - p_h\|_\infty + \|\hat{p}_h - \tilde{p}_h\|_\infty$$

$$< c_{\mathcal{X}^0,\alpha+2\delta} + O\left(\frac{1}{nh^d}\right)$$

$$\le c_{\mathcal{X}^0,\alpha+\delta} - O\left(\frac{\rho_n}{h^d} + \sqrt{\frac{\rho_n\log(1/\delta)}{nh^{2d}}}\right)$$

$$\le c_{\mathcal{X},\alpha}$$

$$\le c_{\mathcal{X}^0,\alpha-\delta} + O\left(\frac{\rho_n}{h^d} + \sqrt{\frac{\rho_n\log(1/\delta)}{nh^{2d}}}\right)$$

$$\le c_{P,\alpha-2\delta}.$$

$\square$

**Corollary 7.** *Suppose Assumption 1, 2, A2 hold.*

*(1) Suppose $nh_n^{-d}\rho_n^2 \to 0$. Then with probability $1 - \alpha - 2\delta$,*

$$p_{h_n}^{-1}[2c_{P,\alpha-2\delta},\infty) \subset \hat{p}_{h_n}^{-1}[c_{\mathcal{X},\alpha},\infty) \subset \operatorname{supp}(P_{h_n}).$$

*(2) Suppose $mh_m^{-d}\rho_m^2 \to 0$. Then with probability $1 - \alpha - 2\delta$,*

$$q_{h_m}^{-1}[2c_{Q,\alpha-2\delta},\infty) \subset \hat{q}_{h_m}^{-1}[c_{\mathcal{Y},\alpha},\infty) \subset \operatorname{supp}(Q_{h_m}).$$

*Proof.* (1) From Lemma 6, $\|\hat{p}_h - p_h\| < c_{\mathcal{X},\alpha} \le c_{P,\alpha-2\delta}$. This implies

$$p_{h_n}^{-1}[2c_{P,\alpha-2\delta},\infty) \subset \hat{p}_{h_n}^{-1}[c_{\mathcal{X},\alpha},\infty) \subset \operatorname{supp}(P_{h_n}).$$

(2) can be proven similarly to (1).

$\square$

*Claim* 8. For a nonnegative measure $\mu$ and sets $A, B, C, D$,

$$\mu(A \cap B) - \mu(C \cap D) \le \mu(A\backslash C) + \mu(B\backslash D).$$

*Proof.*

$$\mu(A \cap B) - \mu(C \cap D) \le \mu((A \cap B)\backslash(C \cap D)) = \mu((A \cap B) \cap (C^{\complement} \cup D^{\complement}))$$
$$= \mu((A \cap B) \cap C^{\complement}) \cup (A \cap B) \cap D^{\complement}))$$
$$\le \mu((A \cap B)\backslash C) + \mu(A \cap B)\backslash D)$$
$$\le \mu(A\backslash C) + \mu(B\backslash D).$$

$\square$

From here, let $P_n$ and $Q_m$ be the empirical measures on $\mathcal{X}$ and $\mathcal{Y}$, respectively, i.e., $P_n = \frac{1}{n}\sum_{i=1}^n \delta_{X_i}$ and $Q_m = \frac{1}{m}\sum_{j=1}^m \delta_{Y_j}$.

**Lemma 9.** *Suppose Assumption 1, 2 hold.*

*(1) Let $A \subset \mathbb{R}^d$. Then with probability $1 - \delta$,*

$$|P_n - P|(A) = o(1).$$

*(2) Let $A_n \subset \mathbb{R}^d$ be a sequence of set satisfying $A_n \to \emptyset$, i.e., $\limsup_n A_n = \emptyset$. Then*

$$P_n(A_n) \to 0 \text{ in probability.}$$

*Proof.* Since $P(A_n) \leq P\left(\bigcup_{i=1}^n A_i\right)$ and $\bigcup_{i=1}^n A_i \to \emptyset$ as well, we can assume that $A_n \downarrow \emptyset$, i.e., $A_n \supset A_{n+1}$ for all $n$ and $\bigcap_{n=1}^\infty A_n = \emptyset$.

(1)

Let $P_n^0$ be the empirical measure on $\mathcal{X}^0$, i.e., $P_n^0 = \frac{1}{n}\sum_{i=1}^n \delta_{X_i}$. By using subgaussian tail bound, with probability $1 - \delta$,

$$\left|P_n^0 - P\right|(A) = O\left(\sqrt{\frac{\log(1/\delta)}{n}}\right).$$

And $\left|P_n - P_n^0\right|(A)$ is expanded as

$$\left|P_n - P_n^0\right|(A) = \frac{1}{n}\sum_{i=1}^n \left|I(X_i \in A) - I(X_i^0 \in A)\right|.$$

Under Assumption 2, $\sum_{i=1}^n I\left(X_i \neq X_i^0\right) \leq n\rho_n$, and hence

$$\left|P_n - P_n^0\right|(A) = \frac{1}{n}\sum_{i=1}^n \left|I(X_i \in A) - I(X_i^0 \in A)\right|$$

$$\leq \frac{1}{n}\sum_{i=1}^n I\left(X_i \neq X_i^0\right) = \rho_n.$$

Therefore, with probability $1 - \delta$,

$$|P_n - P|(A) \leq O\left(\rho_n + \sqrt{\frac{\log(1/\delta)}{n}}\right).$$

(2) Note that $P_n(A_n)$ can be bounded as

$$P_n(A_n) \leq P(A_n) + |P_n(A_n) - P(A_n)|.$$

Fix $\delta > 0$ and $\epsilon > 0$. Since $\lim_n P(A_n) = 0$, we can choose $N_1$ such that $P(A_{N_1}) < \epsilon$, and we can choose $N_2$ such that for all $n \geq N_2$, $\mathbb{P}(|P_n - P|(A_{N_1}) < \epsilon) \geq 1 - \delta$. Then for all $n \geq \max\{N_1, N_2\}$, with probability $1 - \delta$,

$$P_n(A_n) \leq P_n(A_{N_1}) \leq P(A_{N_1}) + |P_n - P|(A_{N_1}) < 2\epsilon.$$

Hence $P_n(A_n) \to 0$ in probability.

$\square$

*Claim* 10. Suppose Assumption 1, 2, A1, A2 hold.

(1)

$$\left|Q_m\left(\hat{p}_{h_n}^{-1}[c_{\mathcal{X}}, \infty) \cap \hat{q}_{h_m}^{-1}[c_{\mathcal{Y}}, \infty)\right) - Q_m\left(\text{supp}(P) \cap \text{supp}(Q)\right)\right| \to 0 \text{ in probability.}$$

(2)

$$\left|Q_m\left(\hat{q}_{h_m}^{-1}[c_{\mathcal{Y}}, \infty)\right) - Q_m\left(\text{supp}(Q)\right)\right| \to 0 \text{ in probability.}$$

(3)

$$Q_m\left(\hat{p}_{h_n}^{-1}[c_{\mathcal{X}}, \infty) \cap \hat{q}_{h_m}^{-1}[c_{\mathcal{Y}}, \infty)\right) \to Q\left(\text{supp}(P)\right) \text{ in probability.}$$

(4)

$$Q_m\left(\hat{q}_{h_m}^{-1}[c_{\mathcal{Y}}, \infty)\right) \to 1 \text{ in probability.}$$

*Proof.* (1)

From Lemma 7, with high probability,
$$Q_m \left( p_{h_n}^{-1}[2c_P, \infty) \cap q_{h_m}^{-1}[2c_Q, \infty) \right) \leq Q_m \left( \hat{p}_{h_n}^{-1}[c_{\mathcal{X}}, \infty) \cap \hat{q}_{h_m}^{-1}[c_{\mathcal{Y}}, \infty) \right)$$
$$\leq Q_m \left( \text{supp}(P_{h_n}) \cap \text{supp}(Q_{h_m}) \right).$$

Then from the first inequality, combining with Claim 8 gives
$$Q_m \left( \hat{p}_{h_n}^{-1}[c_{\mathcal{X}}, \infty) \cap \hat{q}_{h_m}^{-1}[c_{\mathcal{Y}}, \infty) \right) - Q_m \left( \text{supp}(P) \cap \text{supp}(Q) \right)$$
$$\geq Q_m \left( p_{h_n}^{-1}[2c_P, \infty) \cap q_{h_m}^{-1}[2c_Q, \infty) \right) - Q_m \left( \text{supp}(P) \cap \text{supp}(Q) \right)$$
$$\geq - \left( Q_m \left( \text{supp}(P) \backslash p_{h_n}^{-1}[2c_P, \infty) \right) + Q_m \left( \text{supp}(Q) \backslash q_{h_m}^{-1}[2c_Q, \infty) \right) \right).$$

And from the second inequality, combining with Claim 8 gives
$$Q_m \left( \hat{p}_{h_n}^{-1}[c_{\mathcal{X}}, \infty) \cap \hat{q}_{h_m}^{-1}[c_{\mathcal{Y}}, \infty) \right) - Q_m \left( \text{supp}(P) \cap \text{supp}(Q) \right)$$
$$\leq Q_m \left( \text{supp}(P_{h_n}) \cap \text{supp}(Q_{h_m}) \right) - Q_m \left( \text{supp}(P) \cap \text{supp}(Q) \right)$$
$$\leq Q_m \left( \text{supp}(P_{h_n}) \backslash \text{supp}(P) \right) + Q_m \left( \text{supp}(Q_{h_m}) \backslash \text{supp}(Q) \right).$$

And hence
$$\left| Q_m \left( \hat{p}_{h_n}^{-1}[c_{\mathcal{X}}, \infty) \cap \hat{q}_{h_m}^{-1}[c_{\mathcal{Y}}, \infty) \right) - Q_m \left( \text{supp}(P) \cap \text{supp}(Q) \right) \right|$$
$$\leq \max \left\{ Q_m \left( \text{supp}(P) \backslash p_{h_n}^{-1}[2c_P, \infty) \right) + Q_m \left( \text{supp}(Q) \backslash q_{h_m}^{-1}[2c_Q, \infty) \right) \right.$$
$$\left. , Q_m \left( \text{supp}(P_h) \backslash \text{supp}(P) \right) + Q_m \left( \text{supp}(Q_{h_n}) \backslash \text{supp}(Q) \right) \right\}.$$

Now, note that from Lemma 5 implies that for all $x \in \text{supp}(P)$, $\liminf_n p_{h_n}(x) > 0$, so $p_{h_n}(x) > 2c_P$ for large enough $n$. And hence
$$\text{supp}(P) \backslash p_{h_n}^{-1}[2c_P, \infty) \to \emptyset.$$

And similar argument holds for $\text{supp}(Q) \backslash q_{h_m}^{-1}[2c_Q, \infty)$, so $\text{supp}(Q) \backslash q_{h_m}^{-1}[2c_Q, \infty) \to \emptyset$ as well.

Then from Lemma 9,
$$Q_m \left( \text{supp}(P) \backslash p_{h_n}^{-1}[2c_P, \infty) \right) = o_{\mathbb{P}}(1), \qquad Q_m \left( \text{supp}(Q) \backslash q_{h_m}^{-1}[2c_Q, \infty) \right) = o_{\mathbb{P}}(1).$$

Also, since $K$ has compact support, for any $x \notin \text{supp}(P)$, $x \notin \text{supp}(P_{h_n})$ once $h_n < d(x, \text{supp}(P))$. Hence $\text{supp}(P_{h_n}) \backslash \text{supp}(P) \to \emptyset$, and similarly $\text{supp}(Q_{h_m}) \backslash \text{supp}(Q) \to \emptyset$ as well. Then again with Lemma 9,
$$Q_m \left( \text{supp}(P_{h_m}) \backslash \text{supp}(P) \right) = o_{\mathbb{P}}(1), \qquad Q_m \left( \text{supp}(Q_{h_m}) \backslash \text{supp}(Q) \right) = o_{\mathbb{P}}(1).$$

And hence
$$Q_m \left( \hat{p}_{h_n}^{-1}[c_{\mathcal{X}}, \infty) \cap \hat{q}_{h_m}^{-1}[c_{\mathcal{Y}}, \infty) \right) \to Q_m \left( \text{supp}(P) \cap \text{supp}(Q) \right) \text{ in probability.}$$

(2)

This can be done similarly to (1).

(3)

Lemma 9 (1) gives that with probability $1 - \delta$,
$$\left| Q_m \left( \text{supp}(P) \cap \text{supp}(Q) \right) - Q \left( \text{supp}(P) \right) \right| \leq o(1).$$
Hence combining with (1) gives the desired result.

(4)

Lemma 9 (1) gives that with probability $1 - \delta$,
$$\left| Q_m \left( \text{supp}(Q) \right) - 1 \right| \leq o(1).$$
Hence combining with (2) gives the desired result.

$\square$

*Proof of Proposition 1.* Now this is a combination of Claim 10 (1) (2).

$\square$

*Proof of Theorem 2.* Now this is a combination of Claim 10 (3) (4).

$\square$

# E  PHILOSOPHY OF OUR METRIC & PRACTICAL SCENARIOS

## E.1  PHILOSOPHY OF OUR METRIC

All evaluation metrics have different resolutions and properties. Here, we designed our proposed metric with the philosophy of evaluating the performance more conservatively based on (topologically and statistically) certain things. More specifically, in a real situation, there may be outliers in the data or samples we receive, noise may be present, and many other problems may arise due to various other unexpected causes. In these situations, two approaches can be used in the assessment. One is to accept ignorance and use all the data together, the other is to systematically select and exclude as much unreliable information as possible and only use reliable information. We chose the latter because we thought seeing a conservatively consistent result had its own merits (At least we think our approach is worth investigating, showing different aspects that have not been explored before).

## E.2  PRACTICAL SCENARIOS

From this perspective, we present two examples of realistic situations where outliers exist in the data and filtering out them can have a significant impact on proper model analysis and evaluation. With real data, there are many cases where outliers are introduced into the data due to human error (Pleiss et al., 2020; Li et al., 2022). Taking the simplest MNIST as an example, suppose our task of interest is to generate 4. Since image number 7 is included in data set number 4 due to incorrect labeling (see Figure 1 of (Pleiss et al., 2020)), the support of the real data in the feature space can be overestimated by such outliers, leading to an unfair evaluation of generative models (as in Section 5.1.2 and 5.2.3); That is, the sample generated with weird noise may be in the overestimated support, and existing metrics without taking into account the reliability of the support could not penalize this, giving a good score to a poorly performing generator.

A similar but different example is when noise or distortions in the captured data (unfortunately) behave adversely on the feature embedding network used by the current evaluation metrics (as in Section 5.1.2 and 5.2.3); e.g., visually it is the number 7, but it is mismapped near the feature space where there are usually 4 and becomes an outlier. Then the same problem as above may occur. Note that in these simple cases, where the definition of outliers is obvious with enough data, one could easily examine the data and exclude outliers a priori to train a generative model. In the case of more complicated problems such as the medical field (Li et al., 2022), however, it is often not clear how outliers are to be defined. Moreover, because data are often scarce, even outliers are very useful and valuable in practice for training models and extracting features, making it difficult to filter outliers in advance and decide not to use them.

On the other hand, we also provide an example where it is very important to filter out outliers in the generator sample and then evaluate them. To evaluate the generator, samples are generated by sampling from the preset latent space (typically Gaussian). Even after training is complete and the generators' outputs are generally fine, there's a latent area where generators aren't fully trained. Note that latent space sampling may contain samples from regions that the generator does not cover well during training ("unfortunate outlier"). When unfortunate outliers are included, the existing evaluation metrics may underestimate or overestimate the generator's performance than its general performance. (To get around this, it is necessary to try this evaluation several times to statistically stabilize it, but this requires a lot of computation and becomes impractical, especially when the latent space dimension is high.)

Especially considering the evaluation scenario in the middle of training, the above situation is likely to occur due to frequent evaluation, which can interrupt training or lead to wrong conclusions. On the other hand, we can expect that our metric will be more robust against the above problem since it pays more attention to the core (samples that form topologically meaningful structures) generation performance of the model.

# F EXPERIMENTAL DETAILS

## F.1 IMPLEMENTATION DETAILS OF EMBEDDING

We summarize the detailed information of our embedding networks implemented for the experiments. In Figure 2, 3, 4, 5, A2, and 6, `P&R` and `D&C` are computed from the features of ImageNet pre-trained VGG16 (fc2 layer), and `TopP&R` is computed from features placed in $\mathcal{R}^{32}$ with additional random linear projection. In the experiment in Figure 5, the SwAV embedder is additionally considered. We implement ImageNet pre-trained InceptionV3 (fc layer), VGG16 (fc2 layer), and SwAV as embedding networks with random linear projection to 32 dimensional feature space to compare the ranking of GANs in Table 1. The random projection is characterized by preserving the information about distances and homological features defined in the higher dimensional spaces by Johnson Linenstrauss Lemma (Johnson et al., 1986).

## F.2 CHOICE OF CONFIDENCE LEVEL

For the confidence level $\alpha$, we would like to point out that $\alpha$ is not the usual hyperparameter to be tuned: It has a statistical interpretation of the probability or the level of confidence to allow error, noise, etc. The most popular choices are $\alpha = 0.1, 0.05, 0.01$, leading to 90%, 95%, 99% confidence. We used $\alpha = 0.1$ throughout our experiments.

## F.3 ESTIMATION OF BANDWIDTH PARAMETER

As we discussed in section 2, since `TopP&R` estimates the manifold through KDE with kernel bandwidth parameter $h$, we need to approximate it. The estimation techniques for $h$ are as follows: (**a**) a method of selecting $h$ that maximizes the survival time ($S(h)$) or the number of significant homological features ($N(h)$) based on information obtained about persistent homology using the filtration method, (**b**) a method using the median of the k-nearest neighboring distances between features obtained by the balloon estimator (for more details, please refer to Chazal et al. (2017), Wagner et al. (2012), and Terrell & Scott (1992)). Note that, the bandwidth $h$ for all the experiments in this paper are estimated via Balloon Bandwidth Estimator.

For (**a**), following the notation in Section A, let the $i$th homological feature of persistent diagram be $(b_i, d_i)$, then we define its life length as $l_i(h) = d_i - b_i$ at kernel bandwidth $h$. With confidence band $c_\alpha(h)$, we select h that maximizes one of the following two quantities:

$$N(h) = \#\{i : l_i(h) > c_\alpha(h)\}, \ S(h) = \sum_i [l_i(h) - c_\alpha(h)]_+.$$

Note that, we denote the confidence band $c_\alpha$ as $c_\alpha(h)$ considering the kernel bandwidth parameter $h$ of KDE in Algorithm 1.

For (**b**), the balloon bandwidth estimator is defined as bellow:

---

**Algorithm 2** Balloon Bandwidth Estimator

---

1: *# h: bandwidth; KND: kth nearest distance; idx: index*
2: *Given $\mathcal{X} = \{X_1, X_2, \ldots, X_n\}$*
3: **for** *$idx = 1, 2, \ldots, n$* **do**
4:      *# Compute L2 distance between $X_{idx}$ and $\mathcal{X}$*
5:      *$d(X_{idx}, \mathcal{X}) = ||X_{idx} - \mathcal{X}||_2^2$,*
6:      *i.e. $d(X_{idx}, \mathcal{X}) = \{\ldots, d(X_{idx-1}, X_{idx}), d(X_{idx}, X_{idx}), d(X_{idx}, X_{idx+1}), \ldots\}$*
7:      *# Define the kth nearest neighbor distance by sorting in ascending order*
8:      *$KND_{idx} = \text{sort}(d(X_{idx}, \mathcal{X}))[k]$*
9: **end for**
10: *Given kth nearest distance set: $KND = \{KND_1, KND_2, \ldots, KND_n\}$*
11: *# Define the estimated bandwidth $\hat{h}$*
12: *$\hat{h} = \text{median}(KND)$*

---

### F.4 MEAN HAMMING DISTANCE

Hamming distance (HD) (Hamming, 1950) counts the number of items with different ranks between A and B, then measures how much proportion differs in the overall order, i.e. for $A_i \in A$ and $B_i \in B$, $HD(A,B) := \sum_{i=1}^{n} 1\{k : A_i \neq B_i\}/n$ where n is the number of items in list A or B, and k is the number of differently ordered elements. The **mean HD** is calculated as follows to measure the average distances of three ordered lists: Given three ordered lists $A, B,$ and $C$, $\bar{HD} = (HD(A,B) + HD(A,C) + HD(B,C))/3$

### F.5 EXPLICIT VALUES OF BANDWIDTH PARAMETER

Since our metric adaptively reacts to the given samples of $P$ and $Q$, we have two $h$s per experiment. For example, in the translation experiment (Figure 2), there are 13 steps in total, and each time we estimate $h$ for $P$ and $Q$, resulting in a total of 26 $h$s. To show them all at a glance, we have listed all values in one place. We will also provide the code that can reproduce the results in our experiments upon acceptance.

| $\mu$ | $\leftarrow$ shift | | | | | | | | | | shift $\rightarrow$ | |
|---|---|---|---|---|---|---|---|---|---|---|---|---|
| real h | 8.79 | 8.60 | 8.58 | 8.67 | 8.63 | 8.71 | 8.66 | 8.73 | 8.54 | 8.63 | 8.54 | 8.75 | 8.78 |
| fake h | 8.46 | 8.86 | 8.60 | 8.49 | 8.44 | 8.80 | 8.55 | 8.63 | 8.74 | 8.61 | 8.58 | 8.83 | 8.60 |

Table A1: Bandwidth parameters $h$ of distribution shift in Section 5.1.1.

| Steps | 0 | 1 | 2 | 3 | 4 | 5 | 6 |
|---|---|---|---|---|---|---|---|
| real h | 10.5 | 10.6 | 10.5 | 10.7 | 10.5 | 10.8 | 10.6 |
| fake h | 11.0 | 10.2 | 10.0 | 9.82 | 9.57 | 9.29 | 8.78 |

Table A2: Bandwidth parameters $h$ of sequential mode drop in Section 5.1.2.

| Steps | 0 | 1 | 2 | 3 | 4 | 5 | 6 | 7 | 8 | 9 | 10 |
|---|---|---|---|---|---|---|---|---|---|---|---|
| real h | 10.5 | 10.6 | 10.6 | 10.8 | 10.3 | 10.6 | 10.7 | 10.6 | 10.4 | 10.4 | 10.6 |
| fake h | 10.6 | 10.5 | 10.7 | 10.3 | 10.4 | 10.0 | 9.88 | 9.31 | 9.19 | 9.07 | 8.79 |

Table A3: Bandwidth parameters $h$ of simultaneous mode drop in Section 5.1.2.

| | Steps | 0 | 1 | 2 | 3 | 4 | 5 | 6 | 7 | 8 | 9 |
|---|---|---|---|---|---|---|---|---|---|---|---|---|
| Scatter | real h | 8.76 | 8.71 | 8.78 | 8.59 | 8.85 | 8.71 | 8.61 | 8.97 | 9.00 | 8.95 |
| | fake h | 8.61 | 8.76 | 8.63 | 8.93 | 8.46 | 8.72 | 8.67 | 8.68 | 9.07 | 8.88 |
| Swap | real h | 8.65 | 8.44 | 8.76 | 8.62 | 8.55 | 8.75 | 8.96 | 8.59 | 8.53 | 8.74 |
| | fake h | 8.52 | 8.55 | 8.68 | 8.97 | 8.87 | 8.94 | 8.84 | 9.05 | 8.94 | 8.90 |

Table A4: Bandwidth parameters $h$ of non-IID noise perturbation in Section 5.1.3.

| $\Psi$ | $\Psi \downarrow$ | | | | | | | | | | $\Psi \uparrow$ |
|---|---|---|---|---|---|---|---|---|---|---|---|
| real h | 6.26 | 6.41 | 6.31 | 6.23 | 6.06 | 6.24 | 6.20 | 6.61 | 6.39 | 6.28 | 6.26 |
| fake h | 1.04 | 1.67 | 2.36 | 2.90 | 3.41 | 3.93 | 4.31 | 4.90 | 5.51 | 5.60 | 6.46 |

Table A5: Bandwidth parameters $h$ of FFHQ truncation trick in Section 5.2.1.

| Steps | 0 | 1 | 2 | 3 | 4 | 5 | 6 | 7 | 8 | 9 |
|---|---|---|---|---|---|---|---|---|---|---|
| real h | 7.09 | 6.95 | 6.96 | 6.92 | 7.02 | 6.81 | 6.86 | 6.88 | 6.78 | 6.67 |
| fake h | 6.87 | 6.75 | 6.88 | 6.64 | 6.40 | 6.64 | 6.41 | 6.44 | 6.62 | 6.04 |

Table A6: Bandwidth parameters $h$ of CIFAR10 sequential mode drop in Section 5.2.2.

| Steps | 0 | 1 | 2 | 3 | 4 | 5 | 6 | 7 | 8 | 9 | 10 |
|---|---|---|---|---|---|---|---|---|---|---|---|
| real h | 6.78 | 6.95 | 7.05 | 6.62 | 6.85 | 6.51 | 6.94 | 6.91 | 6.80 | 7.00 | 6.78 |
| fake h | 6.75 | 6.69 | 7.14 | 6.68 | 6.75 | 6.84 | 6.61 | 6.68 | 6.50 | 6.32 | 6.17 |

Table A7: Bandwidth parameters $h$ of CIFAR10 simultaneous mode drop in Section 5.2.2.

| | Steps | 0 | 1 | 2 | 3 | 4 | 5 | 6 | 7 | 8 | 9 |
|---|---|---|---|---|---|---|---|---|---|---|---|---|
| Scatter | real h | 7.24 | 7.03 | 7.01 | 7.45 | 7.38 | 7.66 | 7.46 | 7.36 | 7.79 | 7.66 |
| | fake h | 5.41 | 5.77 | 5.67 | 5.49 | 5.71 | 5.65 | 5.82 | 6.08 | 5.81 | 6.02 |
| Swap | real h | 7.25 | 7.20 | 6.99 | 6.67 | 7.07 | 6.96 | 7.03 | 6.94 | 6.83 | 6.86 |
| | fake h | 6.87 | 6.63 | 6.60 | 6.91 | 6.74 | 6.77 | 6.92 | 6.83 | 6.83 | 6.82 |

Table A8: Bandwidth parameters $h$ of FFHQ non-IID noise perturbation in Section 5.2.3.

| | Noise intensity | 0 | 1 | 2 | 3 |
|---|---|---|---|---|---|
| Gaussian noise | real h | 6.52 | 6.27 | 6.41 | 6.49 |
| | fake h | 6.32 | 6.15 | 4.06 | 2.64 |
| Gaussian blur | real h | 6.08 | 6.26 | 6.26 | 6.47 |
| | fake h | 6.12 | 5.13 | 4.60 | 3.40 |
| Black rectangle | real h | 6.06 | 6.28 | 6.49 | 6.44 |
| | fake h | 6.36 | 6.66 | 6.31 | 5.93 |

Table A9: Bandwidth parameters $h$ of FFHQ noise addition in Section 5.2.4.

| | | StyleGAN2 | ReACGAN | BigGAN | PDGAN | ACGAN | WGAN-GP |
|---|---|---|---|---|---|---|---|
| InceptionV3 | real h | 2.605 | 2.629 | 2.581 | 2.587 | 2.615 | 2.606 |
| | fake h | 2.226 | 2.494 | 2.226 | 2.639 | 2.573 | 2.063 |
| VGG16 | real h | 7.732 | 8.021 | 8.175 | 7.792 | 7.845 | 8.036 |
| | fake h | 7.880 | 6.839 | 7.178 | 6.174 | 6.839 | 3.615 |
| SwAV | real h | 0.774 | 0.765 | 0.785 | 0.780 | 0.780 | 0.822 |
| | fake h | 0.740 | 0.667 | 0.6746 | 0.619 | 0.627 | 0.502 |

Table A10: Bandwidth parameters $h$ of GAN ranking in Section 5.2.5.

# G ADDITIONAL EXPERIMENTS

## G.1 TRUCATION TRICK

$\psi$ is a parameter for the truncation trick and is first introduced in Brock et al. (2018) and Karras et al. (2019). We followed the approach in Brock et al. (2018) and Karras et al. (2019). GANs generate images using the noise input $z$, which follows the standard normal distribution $\mathcal{N}(0, I)$ or uniform distribution $\mathcal{U}(-1, 1)$. Suppose GAN inadvertently samples noise outside of distribution, then it is less likely to sample the image from the high density area of the image distribution $p(z)$ defined in the latent space of GAN, which leads to generate an image with artifacts. The truncation trick takes

this into account and uses the following truncated distribution. Let $f$ be the mapping from the input to the latent space. Let $w = f(z)$, and $\bar{w} = \mathbb{E}[f(z)]$, where $z$ is either from $\mathcal{N}(0, I)$ or $\mathcal{U}(-1, 1)$. Then we use $w' = \bar{w} + \psi(w - \bar{w})$ as a truncated latent vector. If the value of $\psi$ increases, then the degree of truncation decreases which makes images have greater diversity but possibly lower fidelity.

## G.2 SURVIVABILITY OF MINORITY SETS IN THE LONG-TAILED DISTRIBUTION

| | Before mode drop | After mode drop |
|---|---|---|
| **Proportion of survival** (before / after filtering) | $100\% \rightarrow 59\%$ | $100\% \rightarrow 57\%$ |
| **TopP&R** (Fidelity / Diversity) | 0.99 / 0.96 | 0.99 / 0.87 (9 p.p $\downarrow$) |
| P&R (Fidelity / Diversity) | 0.73 / 0.73 | 0.74 / 0.70 (3 p.p $\downarrow$) |
| D&C (Fidelity / Diversity) | 0.99 / 0.96 | 1.02 / 0.94 (2 p.p $\downarrow$) |

Table A11: Proportion of surviving minority samples in the long-tailed distribution after the noise exclusion with confidence band $c_\mathcal{X}$. The p.p. indicates the percentage points.

An important point to check in our proposed metric is the possibility that a small part of the total data (i.e., minority sets), but containing important information, can be ignored by the confidence band. We emphasize that since our metric takes topological features into account, even minority sets are not filtered conditioned that they have topologically significant structures. We assume that signals or data that are minority sets have topological structures, but outliers exist far apart and lack a topological structure in general.

To test this, we experimented with CIFAR10, which has 5,000 samples per class. We simulate a dataset with the majority set of six classes (2,000 samples per class, 12,000 total) and the minority set of four classes (500 samples per class, 2,000 total), and an ideal generator that exactly mimics the full data distribution. As shown in the Table A11, the samples in the minority set remained after the filtering process, meaning that the samples were sufficient to form a significant structure. Both `D&C` and `TopP&R` successfully evaluate the distribution for the ideal generator. To check whether our metric reacts to the change in the distribution even with this harsh setting, we also carried out the mode decay experiment. We dropped the samples of the minority set from 500 to 100 per class, which can be interpreted as an 11.3% decrease in diversity relative to the full distribution (Given (1) ratio of the number of samples between majority and minority sets $= 12,000 : 2,000 = 6 : 1$ and (2) 80% decrease in samples per minority class, the true decay in the diversity is calculated as $\frac{1}{(1+6)} \times 0.8 = 11.3\%$ with respect to the enitre samples). Here, recall and coverage react somewhat less sensitively with their reduced diversities as 3 p.p and 2 p.p., respectively, while TopR reacted most similarly (9 p.p.) to the ideal value. In summary, `TopP&R` shows much more sensitiveness to the changes in data distribution like mode decay. Thus, once the minority set has survived the filtering process, our metric is likely to be much more responsive than existing methods.

## G.3 SEQUENTIAL AND SIMULTANEOUS MODE DROPPING WITH CIFAR-10

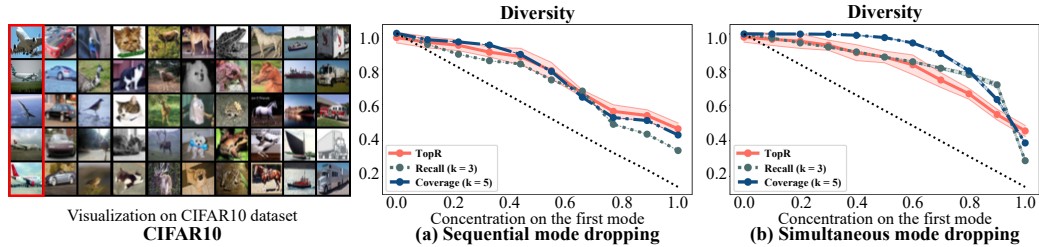

Figure A2: Comparison of evaluation metrics under sequential and simultaneous mode dropping scenario with CIFAR-10.

