# OpenReview forum: "Robust Manifold Estimation Approach for Evaluating Fidelity and Diversity"
_ICLR.cc/2023/Conference — Submitted to ICLR 2023_

### Official Review · Reviewer_xs7v · 2022-10-17

**Confidence:** 5
**Correctness:** 3
**Technical Novelty And Significance:** 2
**Empirical Novelty And Significance:** 3
**Recommendation:** 3

**Clarity, Quality, Novelty And Reproducibility:**

## Clarity

As mentioned above, the current version of the paper is lacking in clarity, in particular when it concerns readers without prior exposure to TDA. Moreover, the original paper on precision and recall by Sajjadi et al. provides a gentler introduction to the required concepts; it might be useful to consider some simple examples presented in this work. Please find some additional comments here:

- The second paragraph of the introduction read more like a separate motivation section for me. Consider rephrasing this.

- I would suggest going into more details for Figure 1 and potentially remove the persistence diagram. To my understanding, the method for estimating confidence bands does *not* make use of topological features directly (of course, they are implicitly being used to create a bound with respect to the Hausdorff distance of functions, but this is a technical detail that can be ignored here); it is thus vital to expand more on their actual utility for this approach.

- What is $p_h^*$ in Section 2.2? I assume this is lifted from one of the articles on persistence diagram confidence bands; please clarify and introduce this.

- "for $\forall$ element" in Algorithm 1 is redundant

- What is $h_n^d$ denoting in Section 3.1? Is this supposed to read $n h^d$?

- Section 5.1.3 is missing a reference to Figure 4, I believe

- What is $\psi$ in Section 5.2.1?

## Quality

The quality of the paper could be introduced in places, in particular when it comes to the experiments:

- Why are no standard deviations of scores being shown? Given a high degree of stochasticity, I would expect this to be the case.

- Why are not other metrics being shown in Figure 8? I would find a comparison with existing metrics to be highly relevant here.

- Adding to this: please disentangle the "outlier removal" step from the calculation of precision and recall. Would it not be possible to subject the measures by Sajjadi et al. to the same treatment?

## Novelty

As mentioned above, I have some trouble assessing the novelty of the paper; a delineation to existing work would be most welcome. Here are some additional comments concerning this aspect:

- When discussing the properties of the new estimator, it appears that existing results are being lifted from literature. This should be made more clear from the beginning; initially, I had some trouble understanding the theoretical contributions of the paper.

## Reproducibility

Since no code is provided, important implementation details are missing and I do not believe that the work is reproducible in its current form (this is also influenced by the clarity issues outlined above).

## Minor suggestions

The paper employs some non-standard and less formal phrasing. Here are some suggestions for improving this:

- I do not understand the phrase "it is of question to to distinguish features [...]". Please rephrase.

- Samples are not "well-curated", but data sets can be well-curated.

- What does "wild" mean for practical scenarios? There is the notion of a "tame" function in persistent homology and computational topology in general; is "wild" supposed to be a contrast to this?

- "tweak" --> "adjust" / "rewrite" / ...

- I would suggest to rephrase "we let the kernel density estimator [...] as" to "we denote the kernel density estimator [...] as". Similar reformulations apply throughout the paper.

- I would rephrase "can have noises" to "can suffer from noise"

- "anagous" --> "analagous"

- "proposedt opological" --> "proposed topological"



**Strength And Weaknesses:**

I see the main **strength** of the paper in making the existing measures more robust towards certain sources of noise and perturbation. Employing a method from topological data analysis is a useful way forward to better characterise the underlying support manifold. The recent years have shown that a better understanding of generative models necessitates also a better understanding of the respective spaces involved; the method at hand provides a useful step in this direction.

While I appreciate the research direction and consider this to be a timely, relevant contribution, the current version of the paper needs to be improved in the following areas:

1. **Clarity (and intuition)**. Even though the method is described as topology-driven, there are almost no explanations of the underlying concepts of topology in the main text. While I am familiar with the cited literature, in particular concerning the reliable estimation of topological features, the paper does not provide sufficient explanations for its approach:
    - Concepts from persistent homology are being used throughout the text but not explained. This will make the paper inaccessible to readers without prior exposure to the theory. I see two ways around this, the first involving a more detailed discussing of topological concepts, the second aiming to rephrase statements in a "less topological" way. For instance, "homological features" could probably be rephrased to "connected components" in this context.
    - It seems to me that only *connected components* are being used to assess the topology. This needs to be clarified, in particular since the persistence diagram shown in Figure 1 works for higher dimensions as well.
    - Likewise, there is no exploitation of multi-resolution structure in the data as far as I can tell. A single threshold is defined for extracting the superlevel set; this is more or less like a "noise suppression technique"; the differences in scale between modes/connected components is not exploited in any way.
    - Given that the main feature of the new method is its purported robustness, more details about this calculation need to be provided. Section 3.2 is too terse for this.
    - Fidelity and diversity as concepts need to be introduced at least briefly.

2. **Novelty and delineation to existing concepts.** Reading the definitions in Section 3.1, my impression is that these are minor modifications of the existing measures, but potentially being calculated in a somewhat simplified space, i.e. the superlevel set of the data with low-persistence (irrelevant) connected components being removed. It is therefore important to provide a better delineation to the existing terms. Moreover, it appears that the main contribution of this paper consists in applying the confidence band calculation method to derive a parameter for removing outliers form an underlying space. If this is the case, would it not be possible to apply this method *as-is* for the "ordinary" definition of precision and recall? In some sense, all the experiments crucially depend on the outlier removal process, so I wonder what would happen if the outlier removal and the calculation of `TopP` and `TopR` would be disentangled.


**Summary Of The Paper:**

This paper contributes a topology-based variant of the precision and recall measures, originally introduced by [Sajjadi et al.](https://proceedings.neurips.cc/paper/2018/file/f7696a9b362ac5a51c3dc8f098b73923-Paper.pdf). The new method makes use of a confidence band analysis, inspired by methods from topological data analysis, to obtain more robust estimates of the underlying support. The proposed measures are thus less affected in the presence of certain outliers and specific types of perturbations. A suite of experiments comparing the new measure with existing methods shows that the proposed method can be gainfully employed in practice.


**Summary Of The Review:**

While I appreciate the direction and subject of this paper, I cannot endorse it for publication in its current form, mainly due to a lack of clarity and intuition, as well as a missing delineation to previous work, which in turns makes it hard to assess the novelty  of the approach. I do believe that filtering outliers using the proposed method is a potential step in the right direction, but in its present form, the paper does not make a compelling case for this. This might change if the topological considerations were to be expanded in order to better discuss their potential benefits.

---

> ### Author Response · Authors · 2022-11-16
> **Comments to R3-8. "Comments on Several Topics"**
>
> **R3-8. "Comments on Several Topics"**
>
> **Note:** Since there are several comments on the different topics, we breakdown this into several pieces to provide an answer by summarizing it here.
>
> ***Clarity.**
>
> **Q1) “What is $p^{*}_{h}$ in Section 2.2?”**
> &ensp; A) The $p^*_{h}$ is a typo and it should be $\hat{p}^*_{h}$. This is the kernel density estimator (KDE) computed on bootstrapped data $\mathcal{X}^*$. This is an issue that arose due to the notation being not unified and we unified our paper notation.
> **Q2) “Section 5.1.3 is missing a reference to Figure 4, I believe”**
> &ensp; A) We added a reference to Figure 4 omitted in section 5.1.3.
> **Q3) “What is $h^{d}_n$ denoting in Section 3.1? Is this supposed to read $nh^d$?”**
> &ensp; A) As in the formal definition of KDE, we assume that the bandwidth $h$ converges to $0$ when the data size $n$ increases. Regarding this the bandwidth $h$ is denoted as $h^{d}_{n}$ showing its dependency to n. We have specified this in our paper.
> **Q4) “What is $\psi$ in Section 5.2.1?”**
> &ensp; A) $\psi$ is a parameter for the truncation trick and is first introduced in [11]. We followed the approach in [11], [12]. GANs generate images using the noise input $z$, which follows the standard normal distribution $\mathcal{N}(0,I)$ or uniform distribution $\mathcal{U}(-1,1)$. Suppose GAN inadvertently samples noise outside of distribution, then it is less likely to sample the image from the high density area of the image distribution $p(z)$ defined in the GAN’s latent space, which leads to generate an image with artifacts. The truncation trick takes this into account and uses the following truncated distribution. Let $f$ be the mapping from the input to the latent space. Let $w=f(z)$, and $\bar{w}=\mathbb{E}[f(z)]$, where $z$ is either from $\mathcal{N}(0,I)$ or $\mathcal{U}(-1,1)$. Then we use $w’=\bar{w}+\psi(w-\bar{w})$ as a truncated latent vector. If the value of $\psi$ increases, then the degree of truncation decreases which makes images have greater diversity but possibly lower fidelity.
>
> &ensp; [11] “Brock, Andrew, Jeff Donahue, and Karen Simonyan. "Large scale GAN training for high fidelity natural image synthesis." arXiv preprint arXiv:1809.11096 (2018).”
> &ensp; [12] Tero Karras, Samuli Laine, and Timo Aila (2019). A style-based generator architecture for generative adversarial networks. https://arxiv.org/abs/1812.04948
> **Q5) “The second paragraph of the introduction … consider rephrasing this.”**
> &ensp; A) At first glance, it might seem that our paper wastes space on providing too much motivation, as you have suggested. However, we intentionally put much motivation, due to that TopP&R lies on the intersection of model evaluation and topological data analysis (TDA), which has been rarely tried before. Potential readers are likely to be rarely familiar with both area of TDA and model evaluation. So emphasizing the motivation is more crucial for those readers to better understand the storyline of our paper. The first paragraph explains importance of the model evaluation, and then the second paragraph motivates to the need of robust and reliable evaluation metric, so this will naturally link to the denoising topological noise via TDA. Hence both paragraphs are essential for motivating to link model evaluation and TDA.
> **Q6) “for \forall element” in Algorithm 1 is redundant.”**
> &ensp; A) Thanks for pointing this out, we fixed it in our main text.
>
> ***Quality.**
>
> **Q1) “Why are no standard deviations of scores … expect this to be the case.”**
> &ensp; A) Thanks for pointing out this. We also agree that it is important to show that our proposed metric works reliably in terms of stochastics. We have repeated the evaluation measurements 10 times for most of the experiments in the paper, and provided their stochasticities with their 1 standard deviations. Thanks to your comment, we were able to significantly improve our manuscript and deepen our understanding of TopP&R even more. We found that TopP&R shows robustness to the noisy features in our updated experiments, which again highlight the reliable characteristics of TopP&R.
> **Q2) “Why are not other metrics being shown in Figure 8?”**
> &ensp; A) Thanks for pointing out the important points. We added F1 scores of PR variant metrics and FID in FIgure 7 (previously Figure 8) of Section 5.2.4. In summary, the result of our metric is consistent with the result of the FID. Our metric captures the subtle difference according to the noise intensity better than the other existing PR variants.

---

> > ### Author Response · Authors · 2022-11-16
> > **Comments to R3-8. "Comments on Several Topics"**
> >
> > ***Novelty) “As mentioned above, I have some trouble … theoretical contributions of the paper”**
> > &ensp; A) Our theoretical results on the consistencies of TopP&R are novel and new. Existing methods for model evaluations are rarely studied under statistical framework, in particular for the literature on the precision and recall such as [7]. [9]. [10]. So there are no direct existing results to be compared, and our theoretical contributions are the first statistical results on model evaluations, as far as we know. Also see our answer to R3-6 for a broader impact of our theoretical contributions to adapting statistical inference under noise to geometrical and topological data analysis.
> >
> > [7] Mehdi Sajjadi, Olivier Bachem, Mario Lucic, Olivier Bousquet, Sylvain Gelly (2018), Assessing Generative Models via Precision and Recall
> > [9] Tuomas Kynkäänniemi, Tero Karras, Samuli Laine, Jaakko Lehtinen, and Timo Aila (2019), Improved precision and recall metric for assessing generative models. https://arxiv.org/abs/1904.06991
> > [10] Muhammad Ferjad Naeem, Seong Joon Oh, Youngjung Uh, Yunjey Choi, and Jaejun Yoo (2020), Reliable fidelity and diversity metrics for generative models. In International Conference on Machine Learning. https://arxiv.org/abs/2002.09797
> >
> > ***Reproducibility) “Since no code is provided, important implementation details … in it s current form”**
> > &ensp; A) Since the code is currently being organized, it will be released right after the final acceptance, and the code that can reproduce the experiments in the paper will also be released.
> >
> > ***Minor suggestions) “The paper employs some non-standard and less formal phrasing. Here are some suggestions for improving this”**
> > &ensp; A) These are good suggestions and we will take them into account. Regarding one of your questions, we used “wild” as a literal word in English describing properties of data in real world practice, but without specifical mathematical meaning. It contains nuance that practical data do not have nicely organized structures or assumptions typically assumed in theoretical analysis. Tameness condition can be one of those assumptions but ‘wild’ is not intended to specifically act as an antonym for tameness. And as you can see from our text, ‘wild’ is placed around paragraphs describing noise and hence it has more nuance in probabilistic assumptions that real world data can be always corrupted with noise.

---

> > > ### Comment · Reviewer_xs7v · 2022-11-16
> > > **On reproducibility**
> > >
> > > > A) Since the code is currently being organized, it will be released right after the final acceptance, and the code that can reproduce the experiments in the paper will also be released.
> > >
> > > This is great! I would still have appreciated an implementation as part of the supplementary materials. While I understand that code cannot be directly shared sometimes, this would have massively helped in the reviewing process, at least from my point of view. As it stands, many of the issues (quality, phrasing, novelty, etc.) could have been addressed by having access to the code (at least as a reviewer).

---

> > > > ### Author Response · Authors · 2022-11-19
> > > > **Author response to "On reproducibility"**
> > > >
> > > > ~~Sorry that we could not share it this time. Given a tight deadline, we spent most of our time preparing the rebuttal so we could not spend much time on the code. However, we will do our best to share it as soon as possible.~~

---

> > > > > ### Author Response · Authors · 2022-11-22
> > > > > **Author response to "reproducibility"**
> > > > >
> > > > > Our code is now available in the link bellow:
> > > > > **https://anonymous.4open.science/r/top_pr-92EC/README.md**

---

> > > > > > ### Comment · Reviewer_xs7v · 2022-11-23
> > > > > > **Thanks for sharing the code!**
> > > > > >
> > > > > > Thanks for sharing the code—already checking it out; I appreciate that!

---

> ### Author Response · Authors · 2022-11-16
> **R3-6 & R3-7. "Reading the definitions in Section 3.1 … connected components being removed. Moreover, it appears that …  apply this method as-is … precision and recall?" & “I would suggest going into more details for Figure 1 … persistence diagram”**
>
>
> **R3-6. “Reading the definitions in Section 3.1 … connected components being removed. Moreover, it appears that …  apply this method as-is … precision and recall?”**
>
> **Point 1)** The proposed method is possibly a simple application of existing PR variants to a method of defining a simplified space that excludes outliers by defining a super-level set of data. If this is the case, this method lacks novelty, as it is a minor modification of existing PR variants.
> **Point 2)** If the existing method is tweaked to be applied in a simplified space based on the confidence band method, then it is likely to be used in a manner similar to the existing PR metric by simply removing the noise filtering procedure.
>
> **Note**: Since the comment is dense and addresses several important issues together, we break it down into several points and address them one by one. The outline of our answer is as follows:
>
> We first explain why TopP&R is not a simple combination of outlier elimination with existing methods.  We clarify why the denoising procedure cannot be separated from our method and show that in fact, separating the two is not useful because it loses topological properties and interpretation. Finally, we also clarify the theoretical novelty of our method.
>
> **A) Why the denoising procedure cannot be disentangled**
> &ensp; As you have mentioned, the core idea of TopP&R is to use bootstrap confidence band to filter out noise and only maintain reliable data. So at first glance, TopP&R seem to be consisting of the outlier removal part and the usual precision recall part, and it seems possible to separate two parts. However, it is not practically feasible to separate them.
> The main difficulty is that there is no single standard way of computing precision and recall from data. For the distributions, it is customary to define the precision and the recall as $Q(\mathrm{supp}(P))$ and $P(\mathrm{supp}(Q))$, respectively, as in Section 3.1 of our paper and in [7]. Then, when we observe $X_{1},...,X_{n}$ from $P$ and $Y_{1},...,Y_{m}$ from $Q$, the most natural sample versions of the precision and the recall should be $Q_{m}(\mathrm{supp}(P_{n}))$ and $P_{n}(\mathrm{supp}(Q_{m}))$, where $P_{n}=\frac{1}{n}\sum_{i=1}^{n} \delta_{X_i}$ and $Q_{m}=\frac{1}{m}\sum_{i=1}^{m} \delta_{Y_j}$ are empirical measures of $X$ and $Y$, respectively. However, when the distributions $P$ and $Q$ are continuous, i.e. having no point mass, then $Q_{m}(\mathrm{supp}(P_{n}))=P_{n}(\mathrm{supp}(Q_{m}))=0$ almost surely, so these natural sample versions don’t recover any useful information of the precision and the recall of the distribution. This is essentially due to that estimating the support of a distribution from sample data is not trivial.
> Hence, to extract useful information of the precision and the recall of the distribution from the data, **different ways of approximating the support** can be done. Essentially speaking, **[7]  used clustering** to approximate the support to a finite set of points, **[9], [10] used k-nearest neighbors** to estimate the support, and **our paper used the super-level set of kernel density estimator (KDE)**. Hence, even after removing outliers, TopP&R are different from existing metrics in approximating the support, and TopP&R are **not mere combinations of outlier removal and existing metrics**.
> And within TopP&R, it is not really practically useful to separate the outlier removal and the precision recall part. This is because the support of KDE is computed using the entire data, and hence the outliers are also used. If we compute the support of KDE without outliers, then we will **lose all the topological properties and interpretations** of our estimated supports. So we have to maintain outliers to estimate the support, so it is **not practically useful** to disentangle outlier removal part from the entire computation of TopP&R.
>
> [9] Tuomas Kynkäänniemi, Tero Karras, Samuli Laine, Jaakko Lehtinen, and Timo Aila (2019), Improved precision and recall metric for assessing generative models. https://arxiv.org/abs/1904.06991
> [10] Muhammad Ferjad Naeem, Seong Joon Oh, Youngjung Uh, Yunjey Choi, and Jaejun Yoo (2020), Reliable fidelity and diversity metrics for generative models. In International Conference on Machine Learning. https://arxiv.org/abs/2002.09797

---

> > ### Author Response · Authors · 2022-11-16
> > **R3-6 & R3-7. "Reading the definitions in Section 3.1 … connected components being removed. Moreover, it appears that …  apply this method as-is … precision and recall?" & “I would suggest going into more details for Figure 1 … persistence diagram”**
> >
> > **Theoretical novelty of proposed method**
> > &nbsp; Consistency is a traditional topic in statistical inference. And consistencies of topological data analysis have been studied in much literature as well. But they are **mostly under i.i.d. setting without considering noise**. Consistencies under noise framework have also been studied in much literature, but **not quite in geometrical or topological setting**. Statistical inference in geometrical or topological setting with noise is in general very rare, for e.g., [5]. Hence our theoretical contribution is mainly on combining consistency of topological analysis and noise framework. Our contribution is not confined to theoretical properties of our TopP&R, but also contributes to finding an appropriate noise framework and adapting statistical inference in geometrical and topological data analysis. We will clearly highlight our theoretical contributions in the main text.
> >
> > [5] Christopher Genovese, Marco Perone-Pacifico, Isabella Verdinelli, Larry Wasserman (2011), Minimax Manifold Estimation.
> >
> > **R3-7. “I would suggest going into more details for Figure 1 … persistence diagram”**
> > &nbsp; A) We agree that the current Figure 1 contains too much detail on topological data analysis given the current level of topological detail in the main text. And as you have mentioned, the confidence band is computed from the $L_{\infty}$ variation of KDE and does not require the persistence diagram of KDE. As you have suggested, due to its conceptual complexity, **we have replaced Figure 1 with** the figure that illustratively explains how our TopP&R work. The old Figure 1 will be moved to more relevant place such as Section B. However, as you can see in the revised figure, we have chosen to slightly retain a topological touch. We think that it is also important to explain the connection between the topological concepts and our metrics through the figure because our motivation, concepts and theoretical properties are closely related to the topological interpretation of the confidence band.

---

> > > ### Comment · Reviewer_xs7v · 2022-11-16
> > > **Thanks**
> > >
> > > Thanks—I appreciate these updates; I think they greatly help in improving the paper! (And I'm writing this as someone that is very partial to TDA methods; I just believe that it's important to make the paper accessible to a wide audience)

---

> > ### Comment · Reviewer_xs7v · 2022-11-16
> > **Clarification**
> >
> > I have one clarification here:
> >
> > > Essentially speaking, [7] used clustering to approximate the support to a finite set of points, [9], [10] used k-nearest neighbors to estimate the support, and our paper used the super-level set of kernel density estimator (KDE). Hence, even after removing outliers, TopP&R are different from existing metrics in approximating the support, and TopP&R are not mere combinations of outlier removal and existing metrics.
> > And within TopP&R, it is not really practically useful to separate the outlier removal and the precision recall part. This is because the support of KDE is computed using the entire data, and hence the outliers are also used. If we compute the support of KDE without outliers, then we will lose all the topological properties and interpretations of our estimated supports. So we have to maintain outliers to estimate the support, so it is not practically useful to disentangle outlier removal part from the entire computation of TopP&R.
> >
> > Sorry for missing something that might appear obvious to the authors, but does this not mean that TopP&R essentially _also_ works on the k-NN level, similar to [9, 10]? That is: if I used the previous versions of P&R but with the extracted connected components of TopP&R, would this not work?

---

> > > ### Author Response · Authors · 2022-11-19
> > > **Author response to "Clarification"**
> > >
> > > Not at all. Thank you for pointing this out. This is certainly an important but subtle thing that our future readers might be wondering as well. We are going to clarify better in the revised manuscript. (We boldlify the points where we would like to emphasize.)
> > > As you have mentioned, you can understand TopP&R as a two step procedure, the outlier removal step and the precision recall evaluation step. TopP&R use confidence band of kernel density estimator (KDE) for the outlier removal step and the super-level set of KDE for evaluation step. And as you have mentioned, one can imagine using KDE confidence band for outlier removal step and k-NN (or any previous versions of P&R) for the evaluation step. This new combination of confidence band + k-NN will still gain some robustness and reliability due to the outlier removal. However, **the estimated support will be different and have a different topological guarantee**. Note that the super-level set of KDE has a guarantee of excluding topological noise due to that those topological noise are measured in the filtration of KDEs and confidence band provides a statistical threshold to the filtration of KDEs for deciding either signal or noise. In contrast, if we switch from super-level set of KDE to k-NN or something else, then the estimated support using k-NN is not a result of the filtration of KDEs, and its topological noise cannot be measured in the filtration of KDEs. As a consequence, confidence band cannot be applied as a statistical threshold for deciding either signal or noise for the k-NN support, and thus it does not have the same guarantee of excluding topological noise anymore. Hence, one can combine our KDE confidence band with k-NN or something else, and still benefit for some robustness and reliability, but then the topological properties that TopP&R have will be lost.

---

> ### Author Response · Authors · 2022-11-16
> **R3-3 & R3-4 & R3-5. “Likewise, there is no exploitation of multi-resolution structure … modes/connected components is not exploited in any way.” & “Given that the main feature of the new method purported robustness, … need to be provided. Section 3.2 is too terse for this.” & “Fidelity and diversity as concepts need to be introduced at least briefly.”**
>
> **R3-3. “Likewise, there is no exploitation of multi-resolution structure … can tell. A single threshold is defined … modes/connected components is not exploited in any way”**
> &nbsp; A) At first glance, our TopP&R appears to only utilize the topology of a single resolution as we estimate the support by the level set at the confidence band c_X and hence the topology at the single resolution c_X is considered. However, multi-resolution structure is also utilized and reflected in TopP&R. “Topological noise” generally corresponds to a topological feature that appears for a small amount of resolutions of data, and “confidence band” quantifies the statistical significance of a topological feature in terms of how often that feature appears in different resolutions. Hence “topological noise” and “confidence band” are inheritably from the multi-resolution structure. And then, due to the choice of c_X, our estimated support filters out our topological noise, so TopP&R also have connections to multi-resolution structure. As you have mentioned, we are not fully extracting all the information of the multi-resolution structure such as the scale differences between modes. However, this does not mean  that TopP&R do not exploit multi-resolution structure at all, nor is such partial utilization of multi-resolution structure topologically meaningless. Topological data analysis is denotatively considered as persistent homology and mapper, but connotatively contains all applications of topology into data analysis, and hence the partial utilization of multi-resolution structure does not detract from our contribution to connecting topological data analysis and model evaluation.
>
> **R3-4. “Given that the main feature of the new method purported robustness, … need to be provided. Section 3.2 is too terse for this”**
> &nbsp; A) In the paper, the robustness via bootstrapping is broken down into Section 2.2, 3.2, and B in the Appendix, so as the reviewer said, Section 3.2may feel too terse. We separated Section 2.2 from the rest because Section 3.2 and B are one of our main contributions, which are new, while Section 2.2 is from existing studies. Sections 2.2 and 3.2 could have been written together to enrich the explanation on the bootstrap bandwidth c_X. But since the contents of Section 2.2 are pre-existing studies, our new interpretation of bootstrapping can be diluted. With this in mind, we think it’s beneficial to describe the contents separately in order to clearly highlight our contribution.
> Also, we separated Section B from Section 3.2 and included it in Appendix because Section B requires understanding  of TDA details. As you have pointed out in R3-1, combining Section 3.2 with Section B in the main text can be overwhelming for prospective readers as it requires a proper understanding of the details in TDA. Hence, we organized the paper so that readers can understand the robustness of TopP&R in a high-level context without knowing details, but the details of TDA are compromised. The details of TDA have been postponed to the Appendix for readers who need a deeper understanding of our work.
>
> **R3-5. “Fidelity and diversity as concepts need to be introduced at least briefly”**
> &nbsp; A) In the context of generative model evaluation, ‘fidelity’ or ‘precision’ is about the ‘accuracy’ of the model, i.e., whether the generated image indeed belongs to the real image distribution or not. It is generally measured by the fraction of the generated data in the real distribution, i.e., $Q(\mathrm{supp}(P))$. On the other hand, ‘diversity’ or ‘recall’ is about the ‘coverage’ of the model, i.e., whether the generative model can reproduce all the real images in the distribution. It is generally measured by the fraction of the real data in the generative distribution, i.e., $P(\mathrm{supp}(Q))$. In [7], these are defined as ‘max precision’ and ‘max recall’ with more details. We will add the brief introduction of these concepts in the main text.
>
> [7] Mehdi Sajjadi, Olivier Bachem, Mario Lucic, Olivier Bousquet, Sylvain Gelly (2018), Assessing Generative Models via Precision and Recall

---

> > ### Comment · Reviewer_xs7v · 2022-11-16
> > **Additional comments**
> >
> > Thanks for answering this queries. I have a general remark about the multi-scale aspect in another comment (which I wanted to be answered in its comment because I think it's important for understanding the paper):
> >
> > > And then, due to the choice of c_X, our estimated support filters out our topological noise, so TopP&R also have connections to multi-resolution structure.
> >
> > This part I understand—but again, this removes some noise _in the connected components_. No higher-order information is being used. Is this correct?
> >
> > I appreciate the updates of the paper and will add further comments after reading it.

---

> > > ### Author Response · Authors · 2022-11-19
> > > **Author response to "Additional comments"**
> > >
> > > It is not only the noisy connected components that are removed; noisy loops and higher-dimensional homologies with their lifetimes below the confidence band are also removed as well. We have updated Section B and Figure A1 (previously Figure 1) with illustrating various cases of topological noises. A connected component (0-dim) with $(birth)\leq c_{X}$ and $(death)\leq c_{X}$ is removed; a connected component (0-dim) with $(birth)\geq c_{X}$ and $(death)\geq c_{X}$ is removed;  a loop (1-dim) with $(birth)\leq c_{X}$ and $(death)\leq c_{X}$ is removed. Hence, not only a connected component outside the estimated support, but one within the estimated support can also be removed. Also, not only a connected component, but a loop can also be removed.

---

> ### Author Response · Authors · 2022-11-16
> **R3-1. & R3-2. “Concepts from persistent homology … throughout the text but not explained” & “It seems to me that only connected components are being used to assess the topology. This needs to be clarified, since … Figure 1 works for higher dimensions as well”**
>
> # Reviewer 3
>
> Thank you for your interest in our work and insightful review. Thankfully, your suggestions gave us an opportunity to improve our paper to be more accessible to readers. Now we will discuss deeper on the details the reviewer asked us.
>
> **R3-1. “Concepts from persistent homology … throughout the text but not explained.”**
> &nbsp; A) As you have commented we used topological concepts without precised definitions or detailed explanations in the main text. We could have either included more details on topological concepts or completely remove them, as you have suggested. However, both directions have their own disadvantages. For the first approach, since prospective readers of this paper are likely to be from the machine learning community without background in topology, loading all the detailed topological concepts could be too overwhelming for the readers. For the second approach, rephrasing without topology will hide the motivation and theoretical properties of our metrics. For example, rephrasing homological features into connected components will restrict our contribution to filtering out only 0-dimensional homological features. Hence we chose the middle option, to still maintain the topological concepts in the main text but at the minimal level, and to postpone the topological details to Section A in appendix. For us it seems not possible to accomplish the desired clarity for all possible readers with different backgrounds, but we will rephrase further for the main text to be more accessible to readers without topological backgrounds, without loading all the detailed topological concepts.
>
> **R3-2. “It seems to me that only connected components are being used to assess the topology. This needs to be clarified, since … Figure 1 works for higher dimensions as well.”**
> &nbsp; A) As you have mentioned, Figure 1 can be misinterpreted as if we are only considering connected components since the data in Figure 1 only have connected components. Also, Section 3.2 might give the impression that only connected components are considered.
> However, we emphasize that our TopP&R utilize not only 0-dimensional homology (connected components), but also 1- and higher-dimensional homologies. This is because the bootstrap confidence band applies to the persistence diagrams of all dimensions simultaneously. For example, in Fig 6-10 in [6], bootstrap confidence bands separate homological signal from noise not only for $0$-dimension, but also for $1$-dimension.
> The current explanation of Section 3.2 is specifically about outliers, so their removal is explained by connected components. We will reinforce this section and discuss impacts on higher-dimensional homologies as well. In addition we moved the previous Figure 1 to Appendix so that we can have a more through discussion with topological concepts there.

---

> > ### Comment · Reviewer_xs7v · 2022-11-16
> > **Questions about persistent homology and usage**
> >
> > Thanks for replying to my comments!
> >
> > I'll use this comment to ask a general question, which appears to be highly relevant at this point:I understand that the confidence band estimation can be used in higher dimensions. However: in the current paper, can you please point out where exactly higher-dimensional features are being used? Moreover, where exactly is the calculation of persistent homology required?
> >
> > **Please correct me if I am wrong**, but to me it seems that the bootstrap procedure is only used to decide upon a general threshold for the selection. Thus, the whole idea of a multi-scale view on the data is currently not utilised.

---

> > > ### Author Response · Authors · 2022-11-19
> > > **Author response to "Questions about persistent homology and usage"**
> > >
> > > We first point out that the actual calculation of persistent homology is only possibly required to choose the parameter $h$ for the kernel density estimator (KDE), which is discussed in Section F.3. If you choose the first method, which is to maximize the survival time $S(h)$ or the number of significant features $N(h)$, then you need to compute the persistent homology for each parameter value $h$ and compare $S(h)$’s or $N(h)$’s. If you choose the second method of using the balloon estimator, no persistent homology is calculated. For the rest of the part, computing persistent homology is not required.
> > >
> > > And to answer your other questions, it depends on your intended meaning of “used” or “utilized”. If you mean whether TopP&R use the computation result of higher-dimensional features or the multi-scale value of data, then “no”, those homological features or the multi-scale are not actually computed for calculating TopP&R. The only topological thing we compute is the confidence band of persistent homology, and since it is the same as the confidence band of KDE, the computation of persistent homology is not required possibly except for choosing the parameter $h$.
> > > However, the higher-dimensional features or the multi-scale value of data are rather implicitly used or utilized. As we have commented in R3-3, “topological noise” and “confidence band” are inheritably from the multi-resolution structure. Presuming the “topological noise” can be only done with the multi-scale point of view of the persistent homology, so topological denoising is itself an application of the multi-scale point of view as well. For the homological features, they are not being utilized as inputs for the next step, but are rather topological interpretations of the super-level set of KDE. And with this perspective, the level of utilizations of 0- and higher-dimensional homologies are the same: the input of the precision and recall evaluation step is the entire super-level set of KDE, not its 0- or higher-dimensional homological features.

---

### Official Review · Reviewer_sgKE · 2022-10-24

**Confidence:** 3
**Correctness:** 3
**Technical Novelty And Significance:** 2
**Empirical Novelty And Significance:** 3
**Recommendation:** 5

**Clarity, Quality, Novelty And Reproducibility:**

Clarity score 3/10: This paper is in general not clearly written. See the Weakness for example.

Quality score 5/10: The proof of consistency seems correct. I haven't checked all the details partly because this paper is hard to follow.

Novelty score 6/10: The proposed metric is relatively new since the idea of using persistence diagram for outlier resistance is natural.

Reproductivity 5/10: There are so many details in the experiments and it seems not easily reproduced without shared codes.

**Strength And Weaknesses:**

Strength:

S1:  A new pair of robust metric for the evaluation of generative models is proposed.

S2: Consistency with robustness is proved for the new metric.

Weakness:

W1: This paper has some parts not clearly written. There are some examples:

(W1.1): The persistence diagram is strange. The x-label and y-label seem incorrect.

(W1.2):In defining precision$_P(\mathcal{Y})$ in Section 3.1, is $Q(supp(Q))$ really needed?

(W1.3): What is $\mathcal{P}$ in "lim inf$_{n\rightarrow \infty}\mathbb{P}\big( \mathcal{P}\in\mathcal{B} \big)$" in Page 13?

(W1.4): How to understand "homological features whose (birth) ≥ $c_X$ and (death) ≤ $c_X$" in Page 13?

(W1.5): In Assumption A1, it says "(3) $K$ is Lipschitz continuous and of second order..". Is this sentence complete?


**Summary Of The Paper:**

This paper proposes a pair of robust metric for the evaluation of generative models, dubbed Topological Precision and Recall (TopP&R). The proposed metric is proved to achieve consistency with robustness. The effectiveness is validated through experiments on both synthetic and real data.

**Summary Of The Review:**

Due to the comments in "Clarity, Quality, Novelty And Reproducibility", I suggest "marginally below the acceptance threshold".


-------------------------After rebuttal---------------------
Many thanks for the authors rebuttal. I read the authors' feedback and other reviewers' comments. Although some of my concerns have been addressed, I am still sorry to suggest "5 borderline reject" due to the "Clarity, Quality, Novelty And Reproducibility".

---

> ### Author Response · Authors · 2022-11-14
> **Comment to R2-3 & R2-4 & R2-5. “(W1.3):What is $\mathcal{P}$ in "$\liminf_{n\rightarrow\infty} \mathbb{P}(\mathcal{P}\in \mathcal{B})$" in Page 13?” & “(W1.5):In Assumption A1, it says "(3) K is Lipschitz continuous and of second order..". Is this sentence complete?” & “There are so many details in the experiments and it seems not easily reproduced without shared codes.”**
>
> **R2-3. “(W1.3): What is $\mathcal{P}$ in "$\liminf_{n\rightarrow\infty} \mathbb{P}(\mathcal{P}\in \mathcal{B})$" in Page 13?”**
>
> A) $\mathcal{P}$ is the persistent homology of the superlevel filtration of the average kernel density estimator $p_{h}=\mathbb{E}[\hat{p}_{h}]$. We will clarify this in the text.
>
> **R2-4. “(W1.5): In Assumption A1, it says "(3) K is Lipschitz continuous and of second order..". Is this sentence complete?”**
>
> A) The additional period(.) was a typo and the sentence is itself complete. Assumption A1 is coming from Assumption 4-6 in [8]. We will rewrite the assumptions with precise mathematical definitions for those who are unfamiliar with them.
>
> [8] Michael Neumann (1998), Strong approximation of density estimators from weakly dependent observations by density estimators from independent observations.
>
> **R2-5. “There are so many details in the experiments and it seems not easily reproduced without shared codes.” (The code implementations)**
>
> A) **Reproducibility**
> ~~Since the code is currently being organized, it will be released right after the final acceptance, and the code that can reproduce the experiments in the paper will also be released.~~
>
> **Details in the experiments (Hyperparameters)**
> As the reviewer said, this is a part that we didn’t write in detail due to the length limit of the text. The hyperparameters we used are the confidence level $\alpha$ and the bandwidth $h$.
> For the confidence level $\alpha$, $\alpha$ is not the usual hyperparameter to be tuned: It has a statistical interpretation of the probability or the level of confidence to allow error, noise, etc. So it depends on the level of confidence you would like in your data. The most popular choices are $\alpha=0.1, 0.05, 0.01$, leading to 90%, 95%, 99% confidence. We used $\alpha = 0.1$ throughout our experiments, and we will specify this in the experiments as well. We added this explanation in **Appendix E.3**.
> Regarding the bandwidth $h$, since our metric adaptively reacts to the given samples of $P$ and $Q$, we have two $h$s per experiment. For example, in the translation experiment (Figure 2), there are 13 steps in total, and each time we estimate $h$ for $P$ and $Q$, resulting in a total of 26 $h$s. To show them all at a glance, we have listed all values in one place in **Appendix E.5**. We will include the meaning of each parameter, its usage, and detailed values used in the revised manuscript. We will also provide the code that can reproduce the results in our experiments.

---

> > ### Author Response · Authors · 2022-11-22
> > **Author response to "reproducibility"**
> >
> > Our code is now available in the link bellow:
> > **https://anonymous.4open.science/r/top_pr-92EC/README.md**

---

> ### Author Response · Authors · 2022-11-14
> **Comment to R2-1 & R2-2. “(W1.1):The persistence diagram is strange. The x-label and y-label seem incorrect.”, “(W1.4):How to understand "homological features whose $(birth) \geq c_{X}$  and $(death) \leq c_{X}$ " in Page 13?” & “(W1.2):In defining $precision_{P}(Y)$ in Section 3.1, is $Q(supp(Q))$ really needed?”**
>
> # Reviewer 2
> Thank you for your detailed review with constructive suggestions. Thanks to your comment, we were able to significantly improve our manuscript and deepen our understanding of TopP&R even more. Below we have responded point by point to your comments.
>
> **R2-1. “(W1.1): The persistence diagram is strange. The x-label and y-label seem incorrect.”, “(W1.4):How to understand "homological features whose $(birth) \geq c_{X}$  and $(death) \leq c_{X}$ " in Page 13?”**
>
> A) We have revised Figure 1 and will clarify this in our revised manuscript (in progress). At first glance, the x-label and y-label are seemed to be swapped, and this is indeed a very common confusion for the persistence diagram from kernel density estimator (KDE). This is since for KDE, the persistence diagram is generated from the ‘super-level’ filtration, not the ‘sub-level’ filtration. For a usual persistence diagram, sub-level filtration is much more common, since when distance-like filtration (Vietoris-Rips complex, alpha complex, distance function, etc) is used, data points are located at low value of the filtration (e.g., distance to data is 0 at data points) so increasing the filtration value is more natural. However, for kernel density estimator (KDE), data points have high KDE values, so ‘decreasing’ the filtration value is more natural, and ‘super-level’ filtration is therefore considered. For this case, the homological feature is born at high value and died at low value, i.e. birth > death. Hence, for the visualization to be comparable to the visualization of persistence diagram from sublevel filtration, it is customary to put death on the x-axis and birth on the y-axis.  And accordingly, homological features at the super-level set at $c_{X}$ correspond to homological features in the persistence diagram whose $(birth)\geq c_{X}$ and $(death) \leq c_{X}$. For more details, refer to Section 4.4 in [6], in particular Figure 5-10.
>
> [6] Brittany Fasy, Fabrizio Lecci, Alessandro Rinaldo, Larry Wasserman, Sivaraman Balakrishnan, Aarti Singh (2014), Confidence sets for persistence diagrams
>
> **R2-2. “(W1.2): In defining $precision_{P}(Y)$ in Section 3.1, is $Q(supp(Q))$ really needed?”**
>
> A) $Q(\mathrm{supp}(Q))$ was appeared as an intermediate term to motivate our definition of TopP. In [7], precision and recall (max precision and max recall in their paper) are just defined as $Q(\mathrm{supp}(P))$ and $P(\mathrm{supp}(Q))$, and our definitions of the distribution version of precision and recall are the same. $Q(\mathrm{supp}(Q))$ is always 1 so this term is not needed if you have full information of Q. However, we are considering the data version, in particular with noise, so neither $P, Q, supp(P), supp(Q)$  is known a priori and needs to be estimated. And when $\mathrm{supp}(P)$ and $\mathrm{supp}(Q)$ are estimated through kernel density estimator (KDE), they are likely to be a bit smaller than $\mathrm{supp}(P)$ or $\mathrm{supp}(Q)$. So $Q(\mathrm{supp}(Q))$ (and $P(\mathrm{supp}(P))$, respectively) is introduced to compensate this bias. In the distribution version ‘$Q(\mathrm{supp}(Q))$’ is always 1, but in the data version ‘$\sum 1(Y_{j} \in \hat{\mathrm{supp}}(Q))$’ is always smaller than 1, compensating the loss in the numerator when defining TopP. So $Q(\mathrm{supp}(Q))$ was appeared as an intermediate term to motivate our definition of TopP (and $P(supp(P))$ for TopR).
>
> [7] Mehdi Sajjadi, Olivier Bachem, Mario Lucic, Olivier Bousquet, Sylvain Gelly (2018), Assessing Generative Models via Precision and Recall

---

### Official Review · Reviewer_YyJA · 2022-10-28

**Confidence:** 4
**Correctness:** 3
**Technical Novelty And Significance:** 2
**Empirical Novelty And Significance:** 2
**Recommendation:** 3

**Clarity, Quality, Novelty And Reproducibility:**

<Clarity> I recommend that the authors improve the writing of the paper. For example, I cannot find which dataset is used to produce the figures in Fig. 4.

<Quality and Novelty> I am not familiar with related works about topological data analysis, so I am not sure about how much the proposed method is novel, but the theoretical results seem to be novel.

<Reproducibility> As 1) the code implementations and 2) important hyperparameters (See <Weaknesses, section 3 above> are not provided, I think reproducibility of the results is difficult.


**Details Of Ethics Concerns:**

Please see <Weaknesses section 1. a> above.

**Strength And Weaknesses:**

<Strengths>

1) The paper addresses an important topic as several metrics have been proposed by the community for evaluating generative models, but there is no global consensus on which metric can better rank the models.

2) The theoretical analysis shows the robustness of the proposed metrics in the presence of outliers / noisy samples. Also, it shows that the proposed metrics are consistent.

3) The experiments demonstrate some degree of noise/outlier robustness in both synthetic and real scenarios.

<Weaknesses>

Although the theoretical insights and the goal of the paper to “develop a metric robust to noise/outliers” are interesting, I think they may have a limited application due to the following points:
In practice, there is neither ground truth about a data point being an outlier nor about the number of modes. One uses the training dataset (real data denoted by X in the paper) to train the generative model and then uses the metrics to measure different aspects of the model like fidelity, diversity, etc. The proposed metric does not provide a metric that can identify noisy/outlier samples before training. Rather, it attempts to filter noisy/outlier samples in (X) and generated images (Y) when quantifying the quality of the trained model.  Now I have the following questions in this regard:
The paper mentions in section 3.1 that: “Using the superlevel set at C_x allows to filter out noise whose KDE values are likely to be small.” However, doing so may result in unwanted consequences. For example, in practical settings, datasets have a long-tailed and imbalanced distribution w.r.t different factors such as age, gender, race, etc. The minority sets in data are not necessarily outliers, but they are likely to have a low KDE value considering that real-world data usually lies on a low-dimensional manifold with disconnected parts in the high-dimensional space [1]. Thus, the proposed metrics may simply discard them and hide the potential problems of the generative model regarding fairness.

I appreciate the experiment in Figs. 2 and 3 that shows precision and recall may not accurately reflect fidelity and diversity in the presence of outliers. However, as I mentioned above, there is no ground truth about a point in data being an outlier in practice, and the generative model gets trained on both inlier and outlier/noisy data. Thus, even if the proposed metric can filter noisy samples in X and Y, its predicted quantities may not reflect the true behavior of the model and its problems.

Based on parts (a) and (b) above, it seems the theorems provided in the paper may also have limited usefulness in practice.
As far as I know, the truncation trick is proposed because the training dataset may not cover all of the latent space of the style GAN. Thus, randomly generating samples from some areas of the latent space of the trained model may result in low-quality/noisy samples. When increasing the diversity in Fig. 5, inevitably, the model will generate low-quality/noisy samples as well which should result in lower precision values. However, TopP remains constant, and the precision score decreases. Does this observation indicate that TopP may be inferior to the regular Precision score in some scenarios?

The proposed method needs several parameters that should be tuned while limited details and no code implementations have been provided. The paper only mentions references about how the values of ‘h’ are calculated for different experiments, but it does not state what those values have been. Also, different choices of ‘alpha’ in Alg. 1 will result in different C_x, which also determines the samples to keep/discard when calculating TopP/TopR. I believe that exploring how the proposed metrics change with different values of alpha/h and suggestions about how to choose them should be provided in the paper.

I think the paper should also consider using self-supervised learned (SSL) representations beyond ImageNet pretrained classifiers to check the relationship between the metrics. It has been shown that SSL representations produce a more reasonable ranking than the latter [3].

[1] Disconnected Manifold Learning for Generative Adversarial Networks, NeurIPS 2018.
[2] On Self-Supervised Image Representations for GAN Evaluation, ICLR 2021.

I would be happy to raise my score if the authors can address my concerns.


**Summary Of The Paper:**

The paper claims that the previous metrics for evaluating generative models (e.g., FID, IS, Precision, and Recall) may not be reliable as they estimate support manifolds based on sample features, and sample features may contain outlier or noisy features. Thus, it proposes a new method for robustly estimating support manifolds using Kernel Density Estimation under topological conditions. The theoretical results show that the proposed metrics are 1) robust to outliers and Non-IID noise and 2) consistent. Finally, the experiments demonstrate relative robustness to outliers and Non-IID perturbations of features for the metrics.

**Summary Of The Review:**

Based on the points above, I think the paper should be enhanced in several aspects:
How such a metric will be useful to develop generative models considering the concerns that I mentioned about the metric may not reveal the problems of the model regarding fairness or the data itself?

Substantial missing details regarding experimental implementations and hyperparameter values should be provided.

The paper should also compare the relationship between different metrics when using self-supervised learned representations as they have been shown to better rank the generative models.

---

> ### Author Response · Authors · 2022-11-14
> **Comment to R1-4 & R1-5 & R1-6. “The proposed method needs several parameters … should be provided in the paper.” & “I am not familiar with related works … to be novel.” & “I recommend that the authors improve … the figures in Fig. 4.”**
>
> **R1-4. “The proposed method needs several parameters that should be tuned … should be provided in the paper.”**
>
> A) **Reproducibility**
> ~~Since the code is currently being organized, it will be released right after the final acceptance, and the code that can reproduce the experiments in the paper will also be released.~~
>
> **Hyperparameters**
> As the reviewer said, this is a part that we didn’t write in detail due to the length limit of the text. The hyperparameters we used are the confidence level $\alpha$ and the bandwidth $h$.
> The confidence level $\alpha$ is not the usual hyperparameter to be tuned: It has a statistical interpretation of the probability or the level of confidence to allow error, noise, etc. So it depends on the level of confidence you would like in your data. The most popular choices are $\alpha=0.1, 0.05, 0.01$, leading to 90%, 95%, 99% confidence. We used $\alpha = 0.1$ throughout our experiments, and we will specify this in the experiments as well. We added this explanation in **Appendix F.2 & F.3**.
> Regarding the bandwidth $h$, since our metric adaptively reacts to the given samples of $P$ and $Q$, we have two $h$s per experiment. For example, in the translation experiment (Figure 2), there are 13 steps in total, and each time we estimate $h$ for $P$ and $Q$, resulting in a total of 26 $h$s. To show them all at a glance, we have listed all values in one place in **Appendix F.5.** We will include the meaning of each parameter, its usage, and detailed values used in the revised manuscript. We will also provide the code that can reproduce the results in our experiments.
>
> **R1-5. “I am not familiar with related works … the theoretical results seem to be novel.”**
>
> A) **Theoretical novelty of proposed method**
> Consistency is the traditional topic in statistical inference. And consistencies of topological data analysis have been studied in much literature as well. But they are **mostly under i.i.d. setting without considering noise**. Consistencies under noise framework have also been studied in much literature but **not quite in geometrical or topological settings**. Statistical inference in geometrical or topological settings with noise is in general very rare, for e.g., [5]. Hence our theoretical contribution is mainly on combining consistency of topological analysis and noise framework. Our contribution is not confined to the theoretical properties of our TopP&R, but also contributes to finding an appropriate nose framework and adapting statistical inference in geometrical and topological data analysis. We will clearly highlight our theoretical contributions in the main text.
>
> [5] Christopher Genovese, Marco Perone-Pacifico, Isabella Verdinelli, Larry Wasserman (2011), Minimax Manifold Estimation.
>
> **R1-6. “I recommend that the authors improve … the figures in Fig. 4.”**
>
> A) Thank you, we have referred Figure 4 in Section 5.1.3. For the writing, we will reorganize the paper by applying the points we addressed so far. Thank you again for your constructive review.

---

> > ### Author Response · Authors · 2022-11-22
> > **Author response to "reproducibility"**
> >
> > Our code is now available in the link bellow:
> > **https://anonymous.4open.science/r/top_pr-92EC/README.md**

---

> ### Author Response · Authors · 2022-11-14
> **Comment to R1-2 & R1-3. “TopP may be inferior to the regular Precision score in some scenarios?” & “I think the paper should also consider using self-supervised learned (SSL) representations … a more reasonable ranking than the latter.”**
>
> **R1-2. “TopP may be inferior to the regular Precision score in some scenarios?”**
> A) As we discussed above, TopP&R pay more attention to the consistent behavior of a model by examining what the model primarily generates, rather than relying on the entire sample, which contains results by chance. From this point of view, the fact that TopP is kept at 1.0 means that StyleGAN2 produces high-quality images most of the time. Thus, this behavior (“TopP remains constant”) does not mean that TopP is inferior to regular precision, but rather reveals its property focusing on **different** perspectives than the other metrics.
> In addition to this, considering the results in Section 5, TopP&R are arguably good metrics that has distinctive merits under various scenarios. Our argument is again supported by the results in Table 1. Here, our metric provides the most consistent generative model rankings across different embeddings (Inception, VGG, SwAV; We have added SwAV results), while agreeing well with the known FID-based rankings between models. All in all, these results imply that TopP&R can serve as a useful, reliable, and consistent assessment indicator for practitioners to develop generative models.
>
> **R1-3. “I think the paper should also consider using self-supervised learned (SSL) representations … a more reasonable ranking than the latter.”**
> A) Thanks for directing us to this interesting paper and thank you for the suggestion.
> We have conducted more experiments with the SwAV embedder in the experiments of Section 5.2.1 and 5.2.5, which again highlight the advantage of our metric.  In Section 5.2.1, additional to VGG16, experiments with SwAV are also done. Here, the trends of TopP and TopR with SwAV are similar to those with VGG16, implying that StyleGAN2 does not produce sufficiently diverse images from SwAV's point of view as well. Also, note that trends of DC with SwAV are quite different from those with VGG16, suggesting that TopP&R produce comparatively consistent evaluations regardless of embedding choices, which is an important virtue as a metric. (Imagine a metric that varies for different embedders that are developed in the future. This is not ideal.)
> Similarly, experiments with SwAV are done in addition to VGG16 and InceptionV3 in Section 5.2.5. The ranking of generative models by F1 scores with SwAV is more consistent with other embedders for TopP&R, compared to other metrics (PR and DC). The ranking of generative models across different embedders and metrics are updated in Table 1, also listed in below table.
>
> **Quantitative Results**
> To quantitatively compare between the similarity of rankings across varying embedders by different metrics, we additionlly computed the Hamming Distance (HD), where where lower HD indicates more similarity, From measuring HD based on Table 1 (above table), TopPR, PR, and DC have HDs of 1.33, 2.66, and 3.0, respectively. From this, TopP&R provides the most consistent ranking across varying embedders. This result is consistent with the consistent evaluation trends in Section 5.2.1.
>
> We conjecture that this is because TopP&R are less affected by different embedders. Loosely speaking, when viewed as an optimization problem, an embedding generally tries to minimize the loss in the high density area. Hence the geometry of high density area is likely to be better preserved compared to outliers or sparse regions. And since TopP&R estimate the support by filtering out low density regions, the estimated support is likely to be stable regardless of embeddings, and hence TopP&R are likely to be stable as well, compared to other metrics estimating the support from full data.
>
> | Rank of models | 1 | 2 | 3 | 4 | 5 | 6 |
> | --------------------- | - | - | - | - | - | - |
> FID | StyleGAN2 | ReACGAN | BigGAN | PDGAN | ACGAN | WGAN  |
> TopPR (VGG16) | StyleGAN2 | BigGAN | ReACGAN | PDGAN | ACGAN | WGAN  |
> TopPR (InceptionV3) | StyleGAN2 | ReACGAN   | BigGAN  | PDGAN  | ACGAN | WGAN  |
> TopPR (SwAV) | StyleGAN2 | BigGAN | ReACGAN | PDGAN  | ACGAN | WGAN  |
> PR (VGG16) | StyleGAN2 | BigGAN | ReACGAN | PDGAN  | ACGAN | WGAN  |
> PR (InceptionV3) | StyleGAN2 | ReACGAN | BigGAN  | PDGAN  | ACGAN | WGAN  |
> PR (SwAV) | StyleGAN2 | BigGAN | ReACGAN | PDGAN  | WGAN  | ACGAN |
> DC (VGG16) | ReACGAN | StyleGAN2 | PDGAN   | BigGAN | ACGAN | WGAN  |
> DC (InceptionV3) | BigGAN | StyleGAN2 | ReACGAN | PDGAN  | ACGAN | WGAN  |
> DC (SwAV) | StyleGAN2 | BigGAN | ReACGAN | PDGAN  | ACGAN | WGAN  |

---

> ### Author Response · Authors · 2022-11-14
> **Comment to R1-1. “Although the theoretical insights … its predicted quantities may not reflect the true behavior of the model and its problems.”**
>
> # Reviewer 1
> Thank you for your detailed review with constructive suggestions. Thanks to your comment, we were able to significantly improve our manuscript and deepen our understanding of TopP&R even more. Especially, we found that TopP&R reacts well to the minority sets and provides the most consistent ranking across varying embedders, which again highlight the reliable characteristics of TopP&R.
>
> **R1-1. “Although the theoretical insights …  its predicted quantities may not reflect the true behavior of the model and its problems.”**
>
> **Note**: Since the comment is dense and addresses several important issues together, we break it down into several points and address them one by one.
>
>   **Point 1)** No ground truth about a point in data being an outlier in practice, and the generative model gets trained on both inlier and outlier/noisy data. Thus, even if the proposed metric can filter noisy samples in X and Y, its predicted quantities may not reflect the true behavior of the model and its problems.
>   **Point 2)** The proposed metric does not provide a metric that can identify noisy/outlier samples before training. Rather, it attempts to filter noisy/outlier samples in (X) and generated images (Y) when quantifying the quality of the trained model.
>   **Point 3)** Unwanted consequences in practical settings, e.g., datasets with a long-tailed and imbalanced distribution; The proposed metrics may simply discard the minority sets in data although they are not necessarily outliers and hide the potential problems of the generative model.
>
> **The outline of our answer is as follows**:
> (Point 1&2) we will first explain the philosophy of our metric and show some reasons why our topological filtering is actually beneficial in practice.
> (Point 3) Then we show that our metric actually works well for the very case (minority set) that the reviewer was concerned with.
>
> A) **Philosophy of our metric**
> We agree with R1’s view that there is no ground truth in practice. Indeed, we never know a priori the category and the quality of our data in practice. In fact, we want to emphasize that we proposed TopP&R metric precisely because of what R1 pointed out. All evaluation metrics have different resolutions and properties. Here, we designed our proposed metric with the philosophy of evaluating the performance more conservatively based on (topologically and statistically) certain things. More specifically, in a real situation, there may be outliers in the data or samples we receive, noise may be present, and many other problems may arise due to various other unexpected causes. In these situations, two approaches can be used in the assessment. One is to accept ignorance and use all the data together as R1 suggested, the other is to systematically select and exclude as much unreliable information as possible and only use reliable information. We chose the latter because we thought seeing a conservatively consistent result had its own merits. (At least we think our approach is worth investigating, showing different aspects that have not been explored before.)

---

> > ### Author Response · Authors · 2022-11-14
> > **Comment to R1-1. “Although the theoretical insights … its predicted quantities may not reflect the true behavior of the model and its problems.”**
> >
> > **Practical scenarios**
> > From this perspective, we present two examples of realistic situations where outliers exist in the data, and filtering out them can have a significant impact on proper model analysis and evaluation.
> > 1. With real data, there are many cases where outliers are introduced into the data due to human error [1,2]. Taking the simplest MNIST as an example, suppose our task of interest is to generate 4. Since image number 7 is included in data set number 4 due to incorrect labeling (see Figure 1 in [1]), the support of the real data in the feature space can be overestimated by such outliers, leading to an unfair evaluation of generative models (see our experiment in Section 5.1.1, 5.1.3, and 5.2.3); That is, the sample generated with weird noise may be in the overestimated support, and existing metrics without taking into account the reliability of the support could not penalize this, giving a good score to a poorly performing generator.
> > 2. A similar but different example is when noise or distortions in the captured data (unfortunately) behave adversely on the feature embedding network used by the current evaluation metrics  (see Section 5.1.3 and Section 5.2.3); e.g., visually it is the number 7, but it is mismapped near the feature space where there are usually 4 and becomes an outlier. Then the same problem as above may occur.
> >
> > Note that in these simple cases, where the definition of outliers is obvious with enough data, one could easily examine the data and exclude outliers a priori to train a generative model. In the case of more complicated problems such as the medical field [2], however, it is often not clear how outliers are to be defined. Moreover, because data are often scarce, even outliers are very useful and valuable in practice for training models and extracting features, making it difficult to filter outliers in advance and decide not to use them.
> >
> > [1] Pleiss, Geoff, et al. "Identifying mislabeled data using the area under the margin ranking." Advances in Neural Information Processing Systems 33 (2020): 17044-17056.
> > [2] Li, Zhengwen, Runmin Wu, and Tao Gan. "Study on image data cleaning method of early esophageal cancer based on VGG_NIN neural network." Scientific Reports 12.1 (2022): 1-10.
> >
> > On the other hand, we also provide an example where it is very important to filter out outliers in the generator sample and then evaluate them. To evaluate the generator, samples are generated by sampling from the preset latent space (typically Gaussian). As R1 pointed out, even after training is complete and the generators’ outputs are generally fine, there's a latent area where generators aren't fully trained. Note that latent space sampling may contain samples from regions that the generator does not cover well during training ("unfortunate outlier"). When unfortunate outliers are included, the existing evaluation metrics may underestimate or overestimate the generator's performance than its general performance. (To get around this, it is necessary to try this evaluation several times to statistically stabilize it, but this requires a lot of computation and becomes impractical, especially when the latent space dimension is high.)
> >
> > Especially considering the evaluation scenario in the middle of training, the above situation is likely to occur due to frequent evaluation, which can interrupt training or lead to wrong conclusions. On the other hand, we can expect that our metric will be more robust against the above problem since it pays more attention to the core (samples that form topologically meaningful structures) generation performance of the model.

---

> > > ### Author Response · Authors · 2022-11-14
> > > **Comment to R1-1. “Although the theoretical insights … its predicted quantities may not reflect the true behavior of the model and its problems.”**
> > >
> > > **Dataset with long-tailed distribution having minority sets**
> > > This is a very good point. As mentioned in R1's second comment, real data may have long tails and the evaluation metric should be able to account for this as well. We guess that R1 probably thought that because of our filtering process with a confidence band, TopP&R would drop all minority sets, so the model's performance could not be properly assessed. R1's concerns are totally valid and we agree that it is important to clarify this point.
> > >
> > > We want to emphasize that since our metric takes topological features into account, even minority sets are not filtered conditioned that they have topologically significant structures. Here, we assume that the minority set and the outliers have different topological properties. In other words, we assume that signals or data that are minority sets have topological structures, but outliers exist far apart and lack a topological structure in general. Of course, there may be a case where the number of samples in the minority set is extremely small, making it difficult to examine whether it is the signal or noise. In this case, in the same context as in the answer to the first comment, we cannot say whether the data is a signal or not, so we do not use it and choose the direction of the assessment based on more reliable data.
> > >
> > > We emphasize that our metric is much more sensitive to changes in data distribution like mode decay (**Appendix G.2**). Thus, once the minority set has survived the filtering process, our metric is likely to be much more responsive than existing methods, despite R1’s concerns.
> > >
> > > ***Experimental results on simulating minority sets:** To see whether TopP&R work well on minority sets, it is necessary to test whether the minorities actually survive in practice. To test this, we experimented with CIFAR10, which has 5000 samples per class. We simulate a dataset with the majority set of six classes (2,000 samples per class, 12,000 total) and the minority set of four classes (500 samples per class, 2,000 total), and an ideal generator that exactly mimics the full data distribution. As shown in the table below, the samples in the minority set remained after the filtering process (100% -> 59%), meaning that the samples were sufficient to form a significant structure.
> > > Both D&C and TopP&R successfully evaluate the distribution for the ideal generator. To check whether our metric reacts to the change in the distribution even with this harsh setting, we also carried out the mode decay experiment. We dropped the samples of the minority set from 500 to 100 per class, which can be interpreted as an 11.3 % decrease in diversity relative to the full distribution (# of samples in majority : minority = 12,000: 2,000 = 6:1 & 80% decrease per minority class -> 1/(1+6)*0.8 = 11.3% with respect to the entire samples). Here, recall and coverage react somewhat less sensitively with their reduced diversities as 3 p.p. and 2 p.p., respectively, while TopR reacted most similarly (9 p.p.) to the ideal value. (Here, p.p. means percentage points.)
> > > |  | Before mode drop | After mode drop |
> > > | --------| -------------------------| --------------------- |
> > > | Proportion of survival in the minority set (before / after filtering) | 100% -> 59% | 100% -> 57% |
> > > | TopP&R (Fidelity / Diversity) | 0.99 / 0.96 | 0.99 / 0.87 |
> > > | P&R (Fidelity / Diversity) | 0.73 / 0.73 | 0.74 / 0.70 |
> > > | D&C (Fidelity / Diversity) | 0.99 / 0.96 | 1.02 / 0.94 |
> > >
> > > ***Minor correction:**  “the minority sets in data are likely to have a low KDE value considering that real-world data usually lies on a low-dimensional manifold with disconnected parts in the high-dimensional space.”
> > > A) Finally, we would like to point out a small but common misconception in KDE. Unlike the common belief, in general, data lying on a low-dimensional manifold have **high** KDE values. When data are supported on a $k$-dimensional manifold in $\mathbb{R}^d$, the probability of data generating distribution at the support $x$ grows on a ball as $P(B(x,r))\sim r^k$, where $B(x,r)$ is a ball centered at $x$ and radius $r$. Refer to Lemma 25 in [3] or Lemma 6 in [4]  for more details. In this case, the usual KDE $\hat{p}(x)=\frac{1}{nh^d} \sum K((x-X_i)/h)$ blows up in the order of $h^{k-d}$ when $h\rightarrow 0$. Refer to Theorem 7 in Jiang (2017) for more details (in this paper, the scale of KDE is modified as $\tilde{p}(x)=\frac{1}{nh^k} \sum K((x-X_i)/h)$ to have proper convergence).
> > >
> > > [3] Jisu Kim, Jaehyeok Shin, Alessandro Rinaldo, Larry Wasserman (2019), Uniform Convergence Rate of the Kernel Density Estimator Adaptive to Intrinsic Volume Dimension. https://arxiv.org/abs/1810.05935
> > > [4] Heinrich Jiang (2017), Uniform Convergence Rates for Kernel Density Estimation. https://proceedings.mlr.press/v70/jiang17b.html

---

### Author Response · Authors · 2022-11-14
**Paper revision plan & General comments**

# Paper revision plan
First revision of our paper has been uploaded. A partial list of experiments have been completed and the paper will be revised accordingly. We'd like to share our paper revision and experiment plans :

First revision (**Updated**)
- Overview for clarity (All reviewers): **Figure 1**
- Clarify the philosophy and the theoretical contributions of our metric (All reviewers): **highlighted in "lime"**
- Test on the dataset with minority sets (R1): **highlighted "lime" in Appendix G.2**
- Test with another embedding network (SwAV) (R1): **Figure 5 (section 5.2.1), Table 1 (section 5.2.5)**
- Details of our experimental setting, hyperparameters, and reproducibility (all reviewers):  **highlighted in “lime”**
- Standard deviation for the experiments (R3): **Figure 2 (section 5.1.1), Figure 3 (section 5.1.2), Figure 4 (section 5.1.3), Figure A2 (section 5.2.2), Figure 6 (section 5.2.3)**
- Comparison with the existing metrics in **Figure 7** (R3): **updated**

Second revision (**Updated**)
- Detailed response to R3: **updated**
- **Text updates on** “Clarifying the philosophy and the theoretical contributions of our metric (All reviewers)”: **highlighted in "lime"**
- Detailed explanations of topological denoising process, with an illustrative example of various noises and how they are denoised (R3): **Section B, Figure A1** (previously Figure 1)
- Our code is now available in the following link: **https://anonymous.4open.science/r/top_pr-92EC/README.md**

Second revision of our paper has been uploaded. We are pleased to announce that all the experiments that we planned for the rebuttal have been completed and the paper has been revised accordingly. Although Phase 1 of the revision has been completed, we will continue to revise our manuscript to better reflect the points raised during the rebuttal process and to improve our presentation.

# General Comment
We thank all reviewers for thorough reviews and constructive suggestions on our work. We appreciate the acknowledgments: the paper addresses an important topic (R1, R2, R3) in a timely manner (R3), the idea is new (R2) and interesting (R1), and the method provides a useful step for better understanding generative models (R3). We addressed all the raised concerns as shown below.

---

### Decision · Program_Chairs · 2023-01-20

**Decision:**

Reject

**Justification For Why Not Higher Score:**

The paper has clarity issues as pointed out by reviewers, and it needs to be addressed before publication.

**Justification For Why Not Lower Score:**

N/A

**Metareview: Summary, Strengths And Weaknesses:**

In this paper, the authors propose a rubust and reliable evaluation metric for generative models. More specifically, the authors propose a new method for robustly estimating support manifolds using Kernel Density Estimation under topological conditions. The main strength of the paper is that the authors makes the existing measures more robust towards certain sources of noise and perturbation. However, the paper has clarity issues as pointed out by reviewers; it needs a significant revision. Thus, I encourage the authors to revise the paper based on the reviewer's comments and resubmit it to a future venue.

**Summary Of Ac-Reviewer Meeting:**

N/A